# Metabolic modeling reveals a multi-level deregulation of host-microbiome metabolic networks in IBD

Jan Taubenheim [1,8] ✉, A. Samer Kadibalban[1,6,8], Johannes Zimmermann [1,2,7], Claudia Taubenheim[3], Florian Tran [4,5], Stefan Schreiber[4,5], Philip Rosenstiel [4], Konrad Aden [4,5] & Christoph Kaleta [1] ✉

Inflammatory bowel diseases (IBDs) are chronic disorders involving dysregulated immune responses. Despite the role of disrupted host-microbial interaction in the pathophysiology of IBD, the underlying metabolic principles are not fully understood. We densely profiled microbiome, transcriptome and metabolome signatures from longitudinal IBD cohorts before and after advanced drug therapy initiation and reconstructed metabolic models of the gut microbiome and the host intestine to study host-microbiome metabolic cross-talk in the context of inflammation. Here, we identified concomitant changes in metabolic activity across data layers involving NAD, amino acid, one-carbon and phospholipid metabolism. In particular on the host level, elevated tryptophan catabolism depleted circulating tryptophan, thereby impairing NAD biosynthesis. Reduced host transamination reactions disrupted nitrogen homeostasis and polyamine/glutathione metabolism. The suppressed one-carbon cycle in patient tissues altered phospholipid profiles due to limited choline availability. Simultaneously, microbiome metabolic shifts in NAD, amino acid and polyamine metabolism exacerbated these host metabolic imbalances. Leveraging host and microbe metabolic models, we predicted dietary interventions remodeling the microbiome to restore metabolic homeostasis, suggesting novel therapeutic strategies for IBD.

Inflammatory bowel disease (IBD), with its two main entities Crohn's disease (CD) and ulcerative colitis (UC), represents a severe inflammation of the gastrointestinal tract. While UC is limited to the mucosa of the colon, CD manifests as transmural inflammation in the terminal ileum, the perianal region or even the complete digestive tract[1]. IBD presents as recurrent flares of inflammation that ultimately lead to widespread tissue destruction with a drastically increased risk of sequelae such as colitis-associated cancer or the necessity to surgically remove parts of the digestive tract[2,3]. Previous work has uncovered substantial genetic and environmental contributions to disease susceptibility, but the primary causes of IBD are unknown[1,4–6]. Besides a strong involvement of innate and adaptive immunity, dysbiosis of the

[1]Research Group Medical Systems Biology, Institute of Experimental Medicine, Kiel University and University Hospital Schleswig-Holstein, Kiel, Germany. [2]Research Group Evolutionary Ecology and Genetics, Zoological Institute, Kiel University, Kiel, Germany. [3]Clinic for Internal Medicine II, Hematology and Oncology, University Hospital Schleswig-Holstein, Kiel, Germany. [4]Institute of Clinical Molecular Biology, Kiel University and University Hospital Schleswig-Holstein, Kiel, Germany. [5]Department of Internal Medicine I, University Hospital Schleswig-Holstein, Kiel, Germany. [6]Present address: Institute of Clinical Chemistry, University Hospital Schleswig-Holstein, Kiel, Germany. [7]Present address: Cluster of Excellence Balance of the Microverse, Friedrich Schiller University, Jena, Germany. [8]These authors contributed equally: Jan Taubenheim, A. Samer Kadibalban. ✉e-mail: j.taubenheim@iem.uni-kiel.de; c.kaleta@iem.uni-kiel.de

intestinal microbiome is a hallmark of IBD[1,4–7]. However, to which extent intestinal dysbiosis is a contributor to IBD-associated pathomechanisms or a consequence of intestinal inflammation so far remains unclear.

Current therapeutic approaches aim to specifically interfere with key immune pathways (e.g., TNFα, IL-23, JAK-STAT, IL-6) involved in the perpetuation of the mucosal immune response[8]. Although the advent of novel therapies, particularly JAK1/IL-23 inhibition[9], indicates improved disease control, there is still a considerable amount of approximately 40% of patients that do not benefit from any IBD therapy, leaving a substantial therapeutic gap and opportunities for improvement of outcomes[10].

Previously, we have used constraint-based metabolic modeling of the microbiome to investigate IBD-associated changes in model-predicted microbiome metabolic activity and its association with therapy response during anti-inflammatory therapy[11]. We found that IBD microbiomes were depleted of within-microbiome metabolic exchanges and enriched with interactions that contribute to intestinal dysbiosis, in particular for patients not responding to anti-inflammatory therapy. One hallmark of decreased metabolic exchanges was the reduced production of the anti-inflammatory short-chain fatty acid (SCFA) butyrate in non-responding patients which we could confirm using metabolomics[11]. In other studies, microbial community metabolic modeling revealed that dysbiosis is associated with increased amino acid synthesis and reduced sulfur species production in IBD patients[12] as well as with a reduced potential to produce secondary bile acids[13]. Modeling also revealed that even small IBD related perturbations of the microbial community change the hepatic metabolic potential in glutathione turnover and bile acid metabolism[14]. Furthermore, human metabolic models have been used to contextualize IBD-related gene expression profiles and to subgroup patients using network coherence inferred by reaction activity scores[15]. Another study used context-specific modeling to predict metabolic changes in the CD patient-derived organoids and confirmed accuracy of the model prediction by metabolomics, rendering it a suitable tool to understand inflammation-induced metabolic changes in the host[16].

While metabolic modeling has uncovered various disease-associated metabolic alterations in individual tissues and the microbiome in IBD, it remains unclear whether common patterns exist that could reveal primary drivers of pathology across tissues and thereby differentiate causal relationships from consequential associations. Holistic studies investigating interactions of metabolic functions of patients and the microbiome are sparse and often miss functional explanations[17]. To address this knowledge gap, we applied metabolic modeling to two extensively phenotyped IBD cohorts to understand metabolic alterations in the host and the microbiome during inflammation, remission, and response to treatment. We found a high concordance between changes in metabolic activity on the microbiome side and linked pathways on the host side. In particular, we observe a profound loss of metabolic activity both in inflamed tissue and systemically in the host that is strongly tied to changes we modeled in the microbiome. We found a reduced host NAD metabolism and altered tryptophan metabolism in parallel to a reduced microbial nicotinic acid production. Further, we observed intricate links in the amino acid metabolism of the host and the microbiome, resulting in reduced transamination reactions in IBD patients and consequently leading to reduced systemic glutathione production. Finally, we found reduced one-carbon (C1) metabolism with consequences for the lipid homeostasis in the host, which is exacerbated by the reduced synthesis of homocysteine by the microbiome. Using serum metabolomics data, we confirmed several model-based predictions in host metabolic activity tied to the microbiome and thereby identified potential markers of deregulated host-microbiome-co-metabolism in IBD. Finally, using the derived metabolic microbiome models, we identified potential nutritional interventions that could be used to counteract

inflammation-associated changes in the microbiota that are tied to deregulated inflammatory metabolism in the host.

## Results

### Inflammation is associated with reduced within-community metabolic exchange and altered microbiome-host exchange

To understand how metabolism in the microbiome and patients changes during inflammatory flares, we used data from two longitudinal cohorts of northern German IBD patients (see "Methods") with a total of 296, 324, and 565 biopsy, blood and 16S samples from 62 patients with CD or UC, respectively (Supplementary Fig. 1). To analyse the microbial metabolic changes during IBD, we mapped 16S sequencing data to microbial reference genomes of the HRGM collection[18] (see "Methods"). These genomes were used to reconstruct genome-scale metabolic models and model metabolic fluxes within microbial communities in coupling-based (MicrobiomeGS2[11]) and agent-based (BacArena[19]) approaches (Supplementary Figs. 2 and 4, mean model size: 50.1 ± 21.27 bacteria per model, see "Methods" for details). We predicted flux distributions for the bacterial communities and associated individual reaction fluxes with the patients' disease activity scores (HBI/Mayo score), by building linear mixed models and using the patient identifier as a random effect to account for patient-specific effects in our longitudinal cohort. Disease activity score can be considered as a proxy for inflammation and we will use the terms synonymously throughout the text. We identified 185 different bacterial reactions whose fluxes were associated with inflammation (Fig. 1A and Supplementary Datas 1 and 2), enriched in nine pathways (Fig. 1B). The modeling approaches stress different aspects of microbial ecology (MicrobiomeGS2−cooperation, BacArena−competition), hence we detected relatively little overlap between both methods. Most reactions in these pathways displayed reduced fluxes during inflammation, with six pathways involved in the synthesis of NAD, 2-arachidonoylglycerol, nucleotides, teichoic acid, flavins, and tetrapyrroles. The remaining three pathways were involved in the degradation of complex carbohydrates and their fermentation to SCFAs.

Observing changes in microbiome metabolism, we hypothesized that cross-feeding of metabolites between bacteria (bacteria exchange, see "Methods") would differ in phases of high disease activity. We identified ten metabolites with altered cross-feeding patterns during inflammation (Fig. 1C and Supplementary Datas 3 and 4). While lactate cross-feeding increased, all other metabolites showed reduced cross-feeding. These included reduced amylotriose, glucose, and propionate cross-feeding−metabolites related to fermentation and SCFA production. Further, we found similar associations for oxoglutarate and succinate, two metabolites closely related to the tricarboxylic acid cycle, and alanine and aspartate (key amino acids in various pathways). Notably, glucose, succinate, and aspartate are precursors for flavine[20], tetrapyrrole[21], NAD[22], and nucleotide[23] synthesis, respectively. Therefore, the reduced microbial synthetic pathway activity for NAD, 2-arachidonoylglycerol, nucleotides, teichoic acid, flavins, and tetrapyrroles during inflammation (Fig. 1B) may be partly driven by decreased cross-feeding of these key metabolites among microbes (Fig. 1C).

To understand the consequences of microbiome metabolic changes for the host during inflammation (host exchange, see "Methods"), we investigated the association of microbial production and consumption of metabolites with disease activity. We predicted 19 inflammation-dependent metabolites, including well-described metabolites dysregulated in IBD, such as reduced microbial butyrate production[24,25] and deconjugated bile acids (cholate, glycocholate)[26] production (Fig. 1D and Supplementary Datas 5 and 6). Furthermore, we evaluated which taxonomic groups contributed most to the production and consumption of IBD-associated cross-feeding and host-exchanged metabolites (Fig. 1E, F). Clostridia and Bacilli showed the highest contribution

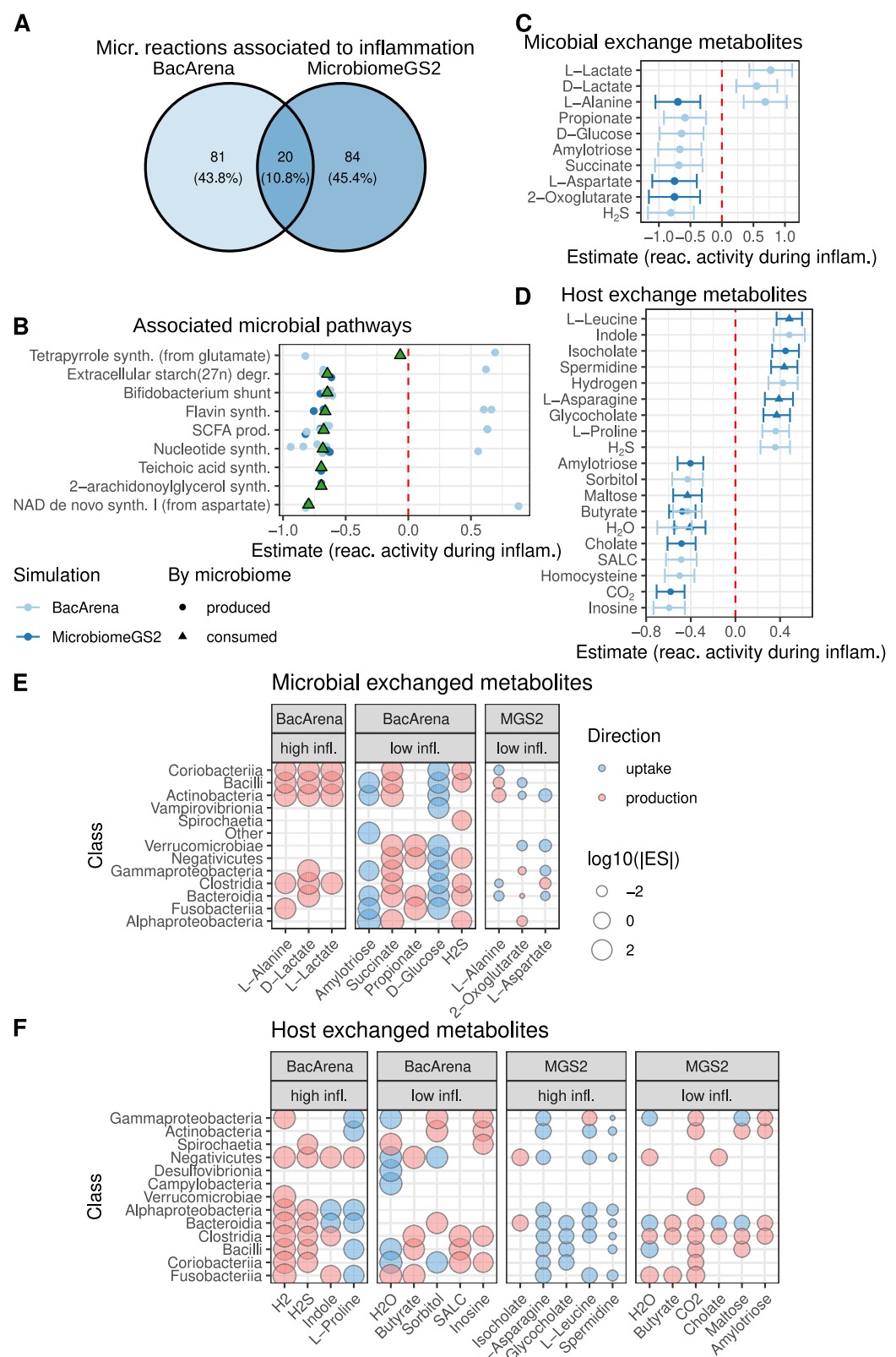

**Fig. 1 | Disease activity-associated microbial metabolism. A** Overlap in inflammation-associated reactions between coupling- and agent-based modeling. **B** Enriched pathways for inflammation-associated reactions. Points represent estimates of linear mixed model association coefficients of significant reaction fluxes with disease activity. Estimates above zero indicate higher activity in inflammation (and vice versa). Triangles express coefficient means. **C, D** Estimates of linear mixed model coefficients for inflammation-associated metabolite exchanges within the microbiota (**C**) and between microbiome and the host (**D**). Centers display the coefficient, whiskers express the confidence intervals of the estimate. Values above zero indicate an upregulation during inflammation. **E, F** Contribution of microbial phylogenetic classes to the uptake/production of target metabolites for within-community exchanges (**E**) and exchanges with the host (**F**). micr. microbial, reac. reaction, infl./inflam. inflammation, MGS2 MicrobiomeGS2, ES effect size (*t*-value), n = 565 samples, multiple testing adjustments via Benjamini–Hochberg correction.

to the cross-feeding of microbiomes (each nine different metabolites, Fig. 1E). Both are producing metabolites whose cross-feeding is associated with high and low inflammation, though low inflammation metabolites are predominant. This indicates that both phyla play important roles in healthy microbiome interactions. Similarly, Bacteroidia predominantly produce metabolites which are associated with low disease activity scores. Additionally, Clostridia exhibited beneficial contributions to the production of health associated metabolites and the consumption of metabolites that are exchanged with the host during inflammation (Fig. 1F).

## Modeling of tissue metabolism in inflammation shows profound changes in lipid and amino acid metabolism

To understand how metabolism of the patients' tissue changes with disease activity, we reconstructed context-specific metabolic models (CSMM, mean model size 5408 ± 716 and 4997 ± 1466 reactions for biopsy and blood, respectively, Supplementary Fig. 2) using bulk RNA isolated from colon biopsies and blood samples and calculated the metabolic potential of these models. We estimated reaction activity using four different modeling-based approaches. First, we directly determined reaction-level expression activities (rxnExpr) using the gene-reaction rules of the models. Second, we considered reaction absence or presence in the CSMM (PA). Third, we used flux variability analysis to determine upper and lower bounds of each reaction which we used to derive centers (FVA.center) and ranges (FVA.range) of fluxes in the CSMMs. We used linear mixed models to test the association of rxnExpr, PA, range.FVA, and center.FVA with disease activity while accounting for patient-specific effects by using patient ID as random factor for each individual reaction. We identified 3115 and 6114 unique reactions significantly associated with disease activity in biopsy and blood samples, respectively (Supplementary Fig. 3A and Supplementary Datas 7 and 8). To facilitate interpretation, we performed an enrichment analysis on the subsystem annotation from Recon3D. We found 25 (24.3%) and 36 (34%) subsystems enriched by these reactions in biopsy and blood samples, respectively (Fig. 2A). Generally, metabolic changes were concordant locally (biopsy) and systemically (blood), with a larger-than-expected overlap in enriched subsystems across tissues (18 subsystems associated in both tissues, $p < 1e^{-4}$, permutation test). Most reactions in these subsystems showed lower activity during inflammation.

For biopsy samples, we observed both an induction and suppression of individual reactions in tryptophan metabolism, starch and sucrose metabolism, and NAD metabolism (Fig. 2A), suggesting a deregulation of these pathways during inflammation.

Inflammation led to notable shifts in modeling-predicted activity in lipid metabolism, including alterations in peroxisomal transport, steroid metabolism, and eicosanoid metabolism. Simultaneously, enzymes associated with fatty acid oxidation, arachidonic acid metabolism, and bile acid metabolism exhibited reduced activity, indicating decreased lipid degradation (Fig. 2A).

To identify key metabolites contributing to the observed metabolic changes, we performed an enrichment analysis for metabolites involved in inflammation-dependent reactions (Fig. 2C, D). We enriched 36 metabolites (20 in biopsies, 25 in blood samples, with nine overlapping) that were more prevalent in our prediction of IBD-associated reactions than would be expected by chance. These "hub metabolites" are expected to have a central role in inflammation-associated metabolism. Almost all reactions linked to the enriched metabolites showed reduced activity during inflammation. Among the common hub metabolites in blood and biopsy samples, we detected central energy metabolites (NAD, FAD), metabolites for general lipid degradation (CoA, carnitine, including SCFAs such as acetate-CoA, malonyl-CoA), and cholesterol. Furthermore, most tissue-specific enriched hub metabolites were also lipid-related: different acyl-CoAs

(biopsy, blood) and cholesterol esters, phosphatidylcholine, and phosphatidylethanolamine (blood). This finding reflects the number of lipid degradation-associated subsystems and underlines the effect of inflammation on lipid metabolism in IBD patients. Among the hub metabolites identified in blood metabolic models, we also found several amino acids (Arg, Gln, His, Lys, Trp), indicating deregulated amino acid metabolism, particularly in the serum of IBD patients.

Our findings in host metabolism integrate well with the disease activity-associated changes in microbiome metabolism. Firstly, many IBD-associated microbial metabolites available to the host are involved in the detected host metabolic subsystems (Fig. 2B and Supplementary Fig. 4A), particularly affecting amino acid metabolic pathways. Secondly, similar metabolic pathways were affected in both, microbiome and patients during inflammation, such as NAD synthesis and metabolism, nucleotide synthesis and interconversion, or reduced synthesis of 2-arachidonoylglycerol and arachidonic/eicosanoid metabolism (Figs. 1B and 2A). Notably, the central energy metabolites NAD and FAD were enriched in inflammation-associated reactions of biopsy and blood samples, aligning with the reduced NAD and flavin synthesis observed in microbial subsystems (Figs. 1B and 2C, D).

## Serum metabolomics confirm metabolic modeling predictions

Next, we aimed to validate the in-silico predictions of host and microbiome metabolism changes by measuring serum metabolite concentrations using targeted metabolomics (*biocrates* MxP® Quant 500). We used linear mixed models to associate serum concentrations with disease activity scores (HBI/Mayo score) in the same manner as before with the predicted metabolic functions (Patient-ID as random factor, HBI/Mayo score as predictor see "Methods", Supplementary Data 9, and Fig. 3).

In association with disease activity, we identified 18 significant metabolites (Fig. 3A and Supplementary Data 10). Nucleotides, sphingolipids, and fatty acids/acylglycerides generally exhibited increased concentrations at higher disease indices, while amino acids and lysophosphatidylcholines were suppressed (Fig. 3A, B).

Sixteen of the 18 metabolites are involved in one or more disease-enriched host subsystems (biopsy: 7, blood: 14, Fig. 3C), confirming our modeling results. High hypoxanthine levels in the blood suggested reduced purine catabolism and nucleotide interconversion during inflammation. Reduced phosphatidylcholines, increased sphingolipids, and fatty acids might result from reduced fatty acid oxidation, glycosphingolipid, and sphingolipid metabolism in the host. Additionally, we observed reduced levels of tryptophan, histidine, and citrulline during inflammation—amino acids and derivatives involved in the urea cycle, tetrahydrobiopterin metabolism, tryptophan metabolism, arginine and proline metabolism, as well as alanine and aspartate metabolism. Notably, for ten of the 14 host subsystems (excluding tetrahydrobiopterin metabolism, sphingolipid metabolism, and glycosphingolipid metabolism) associated with the metabolomic changes, we also identified candidates from significantly altered microbial products during inflammation (Supplementary Fig. 4A). This suggests that the model-predicted changes in microbial metabolism are reflected in the blood metabolome.

## Data layer integration reveals deregulated host microbiome co-metabolic pathways in IBD

To understand potential interactions between microbiome metabolic activity, host metabolic activity, and the host metabolome, we performed network analyses on the host metabolic networks for blood and biopsy. To this end, we constructed a metabolic network with reactions significantly associated with inflammation, only, once for the blood and once for the gut tissue. Onto these networks we mapped IBD-associated microbial and host metabolites (Supplementary Fig. 5 and Supplementary Datas 8, 11, and 12, for more detailed descriptions, see Supplementary Information 1). Since these networks are not flux

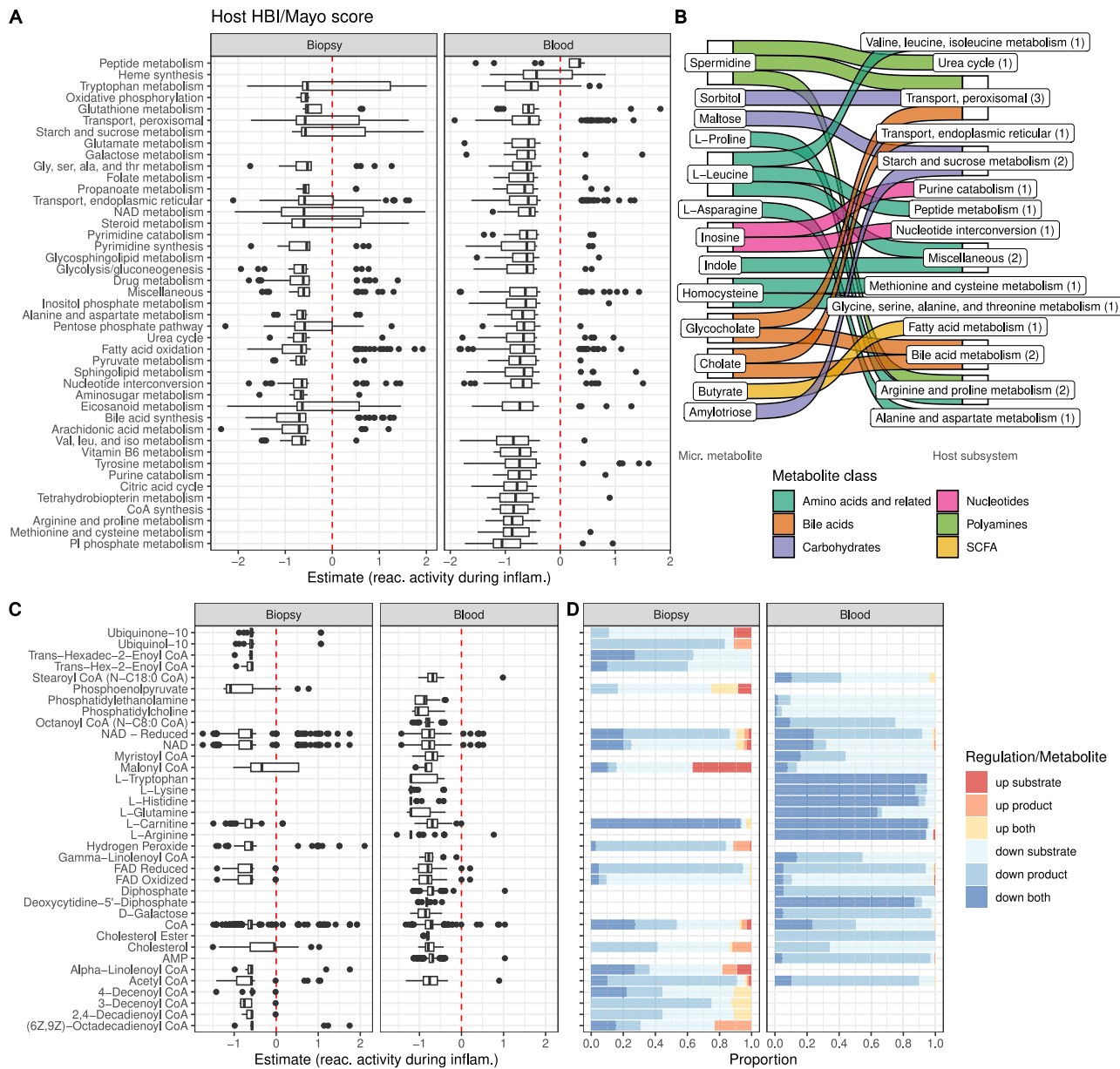

**Fig. 2 | Reduced host metabolic activity during high disease activity.**
**A** Inflammation-association of host reactions grouped into significantly enriched subsystems after linear mixed model association of reaction activity, presence/absence of reaction and reaction range and center to inflammation (see "Methods"). Boxplots represent the estimates of the linear mixed model coefficient to disease activity scores, hence values above zero indicate higher pathway activity during higher inflammation and vice versa. **B** Most IBD-associated microbial metabolites are part of the inflammation-associated host subsystems. **C**, **D** Hub metabolites—metabolites enriched in inflammation-associated reactions. **C** The

boxplot shows estimates of linear mixed models coefficients for association with disease activity scores of reactions which use the hub metabolites. **D** The barplot indicates if metabolites are substrate, product or both in activated (up) or deactivated (down) reactions during inflammation. Boxplots: center line, median; box limits, upper and lower quartiles; whiskers, 1.5 times interquartile range; points, outliers. $n = 296$ for biopsies and $n = 324$ for blood samples, multiple testing adjustments via Benjamini–Hochberg correction. reac. reaction, inflam. inflammation, SCFA short chain fatty acids.

consistent anymore and thus unsuitable for metabolic modeling we fell back to classical network analysis techniques like shortest paths. We used the two networks to understand the relationship of microbial, metabolomics and hub metabolites in terms of path distances (compare Figs. 1D, 2B, and 3A). To this end, we calculated the shortest path (see "Methods") between metabolite pairs of the three categories and compared them to the path length of all other pairs in the network (Fig. 4A–C). A short path was defined as the smallest combined edge length between two metabolites. The edge length was calculated by the sum of the degrees of the connecting metabolites to avoid shortcuts through co-factors in the network (see "Methods")[27]. We found

generally shorter paths for hub and metabolomics metabolites in biopsy samples (Fig. 4A). In blood samples, path lengths between all categories showed shorter paths than the rest of the network (Fig. 4C). This indicates that inflammation-dependent blood metabolism is closely associated with both microbial and metabolomics derived metabolites, whereas gut tissue metabolism appears more dependent on blood metabolites than microbial metabolites.

To further explore the inflammation-dependent networks, we derived closely connected compounds between the different layers. To this end, we determined shortest paths between all pairs of metabolites in the network and sub-selected those pairs of metabolites

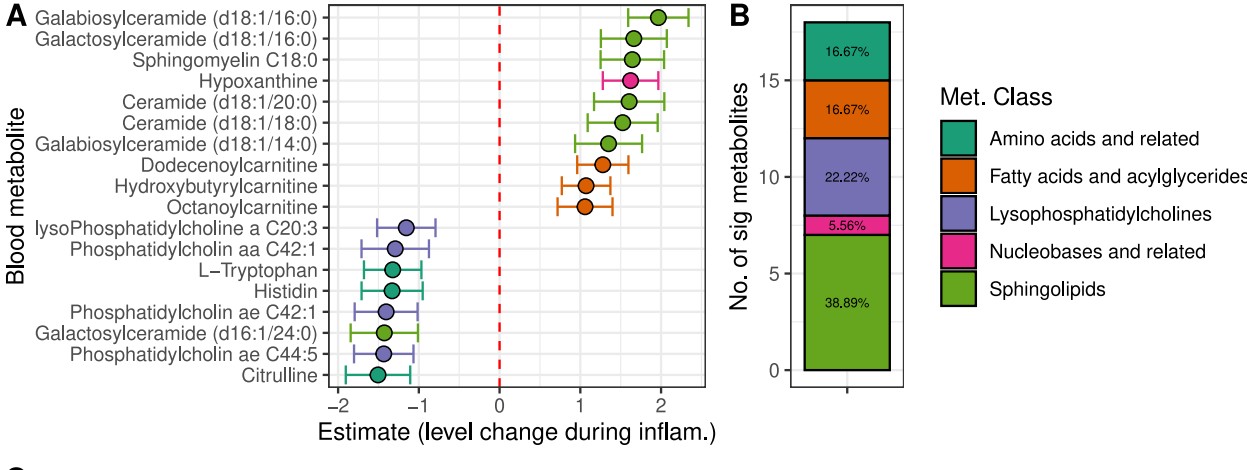

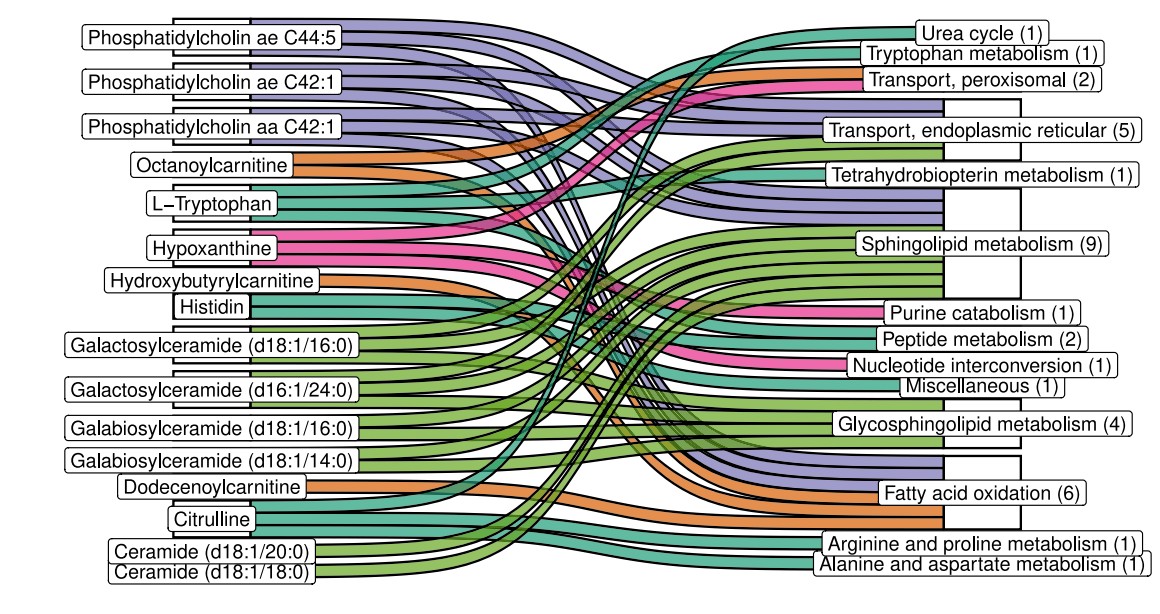

**Fig. 3 | Association between serum metabolomics and disease activity.**
**A** Significant association of serum metabolite concentrations with disease activity scores displayed as estimates for linear mixed model coefficients. Centers display the coefficient, whiskers represent confidence intervals of the linear mixed model (see "Methods"), values above zero indicate higher metabolite levels in high disease activity. **B** Enrichment of inflammation-associated metabolites in metabolite classes. Percentage values indicate the relative abundance of the class.
**C** Connection between inflammation-associated metabolites and modeling-predicted inflammation-associated host pathways. $n = 150$ samples, multiple testing adjustments via Benjamini–Hochberg correction. inflam. inflammation, met. metabolite, no. number.

whose connecting shortest paths were among the shortest 5% (see "Methods" for details) (Fig. 4B, D). In this analysis, we detected numerous connections between microbiome-host metabolism and blood metabolomics related to lipid metabolism. For biopsies, acyl-carnitines (octanoyl-, 3-hydroxybutyryl,- and dodecanoyl-carnitine) were predominantly connected to metabolites of the oxidative phosphorylation pathway, such as FAD, NAD(H), and ubiquinone/ubiquinol (Hub-Met, Fig. 4B). This might explain the increased fatty acid concentrations in blood metabolomics (Fig. 3A) as a consequence of reduced oxidative phosphorylation in gut tissue (Fig. 2A), potentially caused by the reduced SCFA production by the microbiome (Fig. 1D).

Reflecting decreased β-oxidation (Fig. 2A), we detected short paths connecting energy metabolism related hub metabolites with acyl-CoA compounds from the microbiome and the metabolomics (Fig. 4D). Furthermore, several lipids and energy metabolites identified as hub metabolites were connected by downregulated reactions to microbial cholate (Fig. 4D), indicating interdependence of

lipid metabolism of the host and bile acid metabolism of the microbiome.

Further, we observed short paths between different phospholipids and sphingolipids in biopsy samples (Fig. 4B), accompanied by the reduction of their interconversion during inflammation in blood (Fig. 1D) and increasing concentration imbalance of these lipids (Fig. 3A). This hints at a loss of the ability to interconvert these metabolites and balance their concentrations.

To corroborate this notion, we performed a hypergeometric enrichment analysis with the reactions along the shortest paths and identified glycerophospholipid and sphingolipid metabolism as enriched (Supplementary Fig. 6).

Finally, we noticed strong interconnections between amino acids in inflammation-associated metabolism in both tissues and between all groups of metabolites (hub, microbiome, metabolomics). This included paths between asparagine, histidine, and tryptophan, (biopsy, Mic-Met, Fig. 4B) as well as glutamine, arginine, histidine, proline,

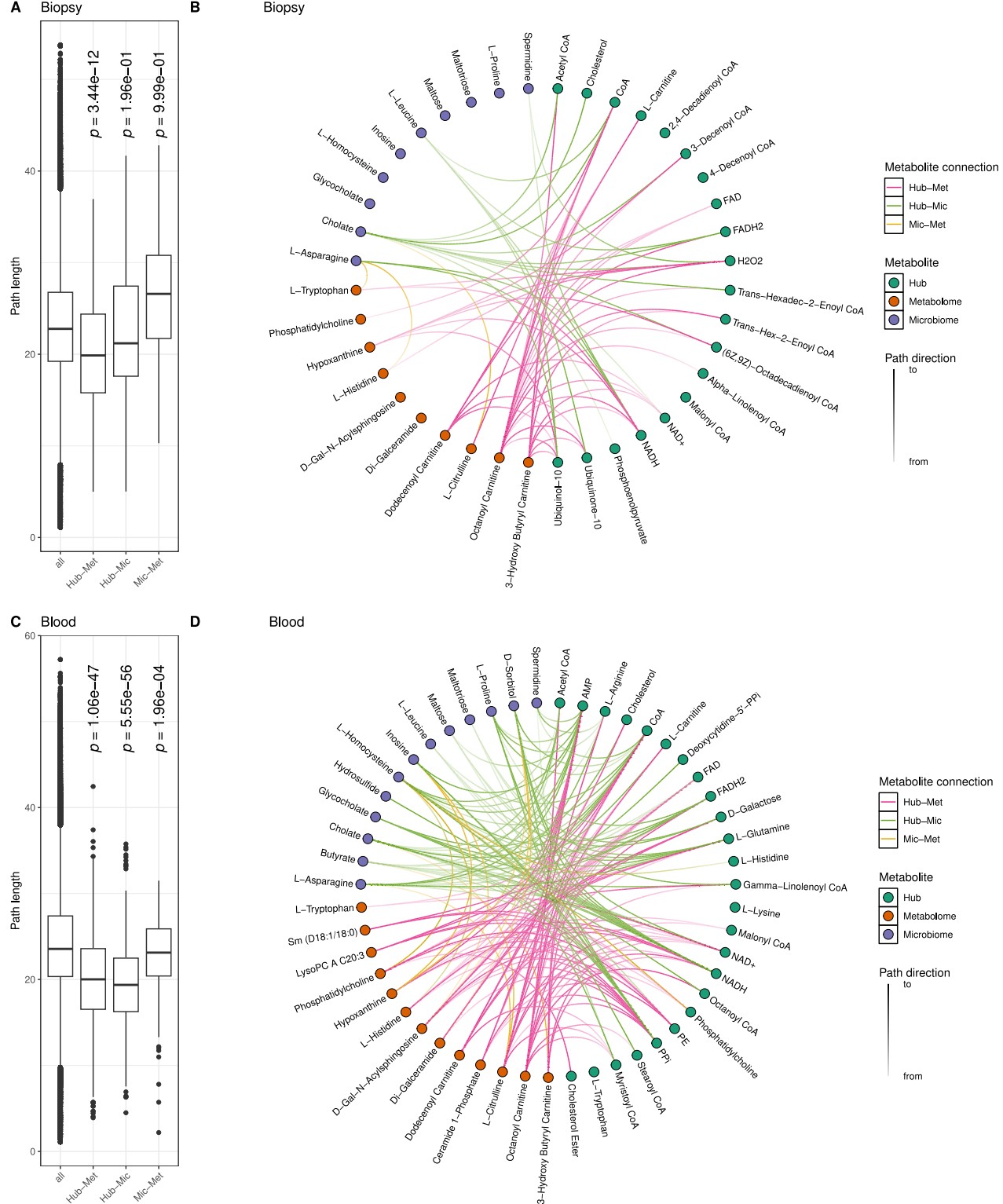

**Fig. 4 | Connected metabolites between microbial-, tissue-metabolism, and metabolomics data layers. A, C** Comparison of path lengths between metabolites from different data layers compared to randomly chosen metabolites. The displayed *p*-value describes results of a one-sided student's *t*-test between the specific group and all other paths (all). **B, D** Connected metabolites between data layers among the 5% shortest paths between all metabolite pairs. Boxplots: center line, median; box limits, upper and lower quartiles; whiskers, 1.5 times interquartile range; points, outliers. For biopsy: all *n* = 1254898, Hub-Met *n* = 260, Hub-Mic *n* = 160, Mic-Met *n* = 49 paths. For blood: all *n* = 2998770, Hub-Met *n* = 496, Hub-Mic *n* = 480, Mic-Met *n* = 213 paths, multiple testing adjustments via Benjamini–Hochberg correction. Hub hub metabolites, Met inflammation-associated metabolites detected in metabolomics, Mic inflammation-associated metabolites detected in microbiome modelling.

asparagine, and leucine (blood, Hub-Met/Hub-Mic, Fig. 4D). Asparagine, glutamine, and arginine form part of a glutamine-arginine axis for amino acid transaminations which prompted us to examine whether amino acid interconversions were particularly affected by inflammation. We searched our data set for reactions facilitating direct interconversion between amino acids (mainly transaminations). Out of the 37 reactions in our data set, we detected 33 reactions (~90%) significantly associated with inflammation (15 for biopsy and 29 for blood, Supplementary Fig. 7). Overall, the findings indicate a central role of amino acid interconversion and $NH_4$ metabolism during inflammation, which was also reflected in the affected host metabolic subsystems, such as decreased glutathione, glutamate, alanine and aspartate, arginine and proline, methionine and cysteine metabolism, and urea cycle (Fig. 2B). Simultaneously, we found aspartate and 2-oxoglutarate (a glutamine derivative) among the metabolites exhibiting decreased exchange among microbes during inflammation (Fig. 1C), emphasizing similar changes in microbial metabolism.

In addition to direct amino acid interconversion, we predicted short paths from homocysteine to arginine, histidine, and phosphatidylethanolamine (Fig. 4D). Conversely, we detected short paths from glutamine and phosphatidylcholine to homocysteine (Fig. 4D). This describes the role of homocysteine in C1 metabolism, which connects amino acid metabolism and membrane lipid metabolism. The enrichment analysis with shortest path reactions confirmed these findings and significantly enriched subsystems for tryptophan, valine, leucine and isoleucine, histidine, alanine and aspartate, arginine and proline, and methionine and cysteine metabolism (Supplementary Fig. 6).

## Disease activity-associated metabolic functions in host and microbiota are attenuated in responders and remitters

So far, we associated metabolic changes with disease activity and unraveled the deregulated parts of the metabolic network, either as a cause or consequence of intestinal inflammation. To better understand the association of metabolism with treatment outcome, we investigated metabolic changes in treatment response ("Response") and remission after treatment ("Remission").

Initially, we scanned our modeling predictions for associations with remission and response at baseline, at 14 days after treatment or with the change between these two time points (Supplementary Fig. 8). We identified only two reactions in the microbial metabolism which were associated with response/remission in this analysis—3-hydroxybutyryl-CoA dehydrogenase and the production of heme (Supplementary Fig. 9).

Next, we investigated associations between modeling predictions and remitter/responder status across all time points except 14 weeks (at which remission is defined based on disease activity). In general, we found fewer significant hits compared to the analysis of the association between modeling predictions in disease activity which, moreover, also mostly overlapped with the results of our previous analysis (see Supplementary Figs. 10–12). While this is essentially expected since remission and response are defined based on changes in disease activity, we wondered whether there is potentially an underlying signature independent of disease activity indicative of therapy response and remission. More specifically, we hypothesized that the deviation of metabolic processes in a patient from the average of the entire patient cohort at a particular state of disease activity could be indicative of overall therapy success. Using linear models, such a test can be performed by determining the association of metabolic processes with remission/response while controlling for disease activity. Indeed, disease activity as a measure of therapy success is strongly influenced by the effect of the therapy itself. Since this effect strongly varies throughout the course of therapy due to changes in dosage and patient-level heterogeneity in response to therapy, controlling for disease activity could therefore uncover the underlying molecular signature of re-establishment of cellular homeostasis independent of the direct effect of therapy.

Determining associations between metabolic activity and response/remitter status while controlling for disease activity, we observed an increase in reaction activity/metabolite levels in responders and remitting patients in particular for those reactions suppressed during inflammation. Thus, we predicted increased NAD and nicotinamide production, increased degradation and fermentation of complex carbohydrates to propionate and butyrate, reduced synthesis and increased consumption of amino acid, and decreased production of diverse lipid species in the microbiome of responding and remitting patients (Supplementary Fig. 6 and Supplementary Information 1).

For the changes in host metabolism, we found increased amino acid metabolism and increased glutathione metabolism in patients responding to treatment (Supplementary Fig. 7C and Supplementary Information 1), reversing some effects observed during inflammation. In patients undergoing remission, we predicted increased sphingolipid metabolism in biopsies, which were reduced during active disease in blood (Supplementary Fig. 7G and Supplementary Information 1). When analyzing metabolite enrichment of the significantly associated reactions with response and remission, we found no consistent patterns between blood and gut tissue (Supplementary Fig. 7D, H and Supplementary Information 1). Notably, for responders, the enrichment of NAD(H) in biopsies and coenzyme A in blood was driven by more upregulated reactions, consistent with the downregulation of these metabolites during inflammation (Supplementary Fig. 7D and Supplementary Information 1).

Association of the metabolomics data with response and remission, revealed an increase of serum levels of lysophosphatidylcholines, phosphatidylcholines, and choline (response only), higher levels of kynurenine and serotonin, and higher levels of cystine (cys-S-S-cys, Supplementary Fig. 13A, D)—coherent with similar negatively associated signals during inflammation (Supplementary Information 1).

In summary, correcting for HBI/Mayo score in the association of metabolic functions to therapy outcome revealed that remission and response patients have a general increase in amino acid, glutathione, sphingolipid, and NAD metabolism. At the same time, activity of these metabolic pathways are reduced during inflammation making them ideal targets for potential therapeutic intervention.

## Inflammation-associated metabolic exchanges are potentially modifiable by dietary interventions

After identifying 59 microbial metabolites whose exchange within the microbiota or between microbiota and host was associated with disease activity, we aimed to determine which of these metabolic exchanges are modifiable through in-silico dietary interventions. This approach could help identify targets for in-vitro testing of dietary supplements that could reverse disease-associated microbial metabolic functions. (Fig. 1C, D, Supplementary Fig. 14C, D, E, H, and Supplementary Datas 13 and 14). First, we excluded metabolites that showed inconsistency between the two community simulation methods that we used (BacArena and MicrobiomeGS2) or across the three disease phenotypes considered in our analysis (inflammation scores, remission and response), that is, metabolites that showed a positive association with inflammation using one method and a negative association using another. We identified 48 metabolic exchange fluxes associated with one or more inflammation measures (intervention targets). Of these, 31 were within-community exchanges, and 17 exchanges between the community and the host.

In order to find the modifiable metabolites among the 48 targets, we tested a total of 221 dietary interventions, i.e., all metabolites that are found in the diet and can be taken up or secreted by the microbiome models. In each intervention, a single dietary metabolite was either removed or its concentration doubled. This resulted in a total of 442 dietary metabolic interventions that were simulated for each of the 476 microbial communities (see "Methods"). We conducted a paired Wilcoxon signed-rank test to examine the impact of dietary

interventions on metabolic fluxes across 490 metabolites (245 fluxes within the community and 245 fluxes with the host). Our analysis revealed that 60 fluxes with the host (24%) and 47 fluxes within the community (19%) were significantly changed as a result of one or more interventions. Out of the 48 inflammation-associated fluxes for target metabolites, 39 could be altered by at least one intervention (23 with the host and 16 within the microbiome). We next tested whether the interventions had the desired effect on metabolic exchanges, for instance, a reduction in flux in case of a positive association with disease activity or an increased flux in case of a negative association. All interventions showed desired and undesired effects on the target metabolic exchanges (Fig. 5). The removal interventions generally had a more pronounced influence on the target metabolic fluxes than doubling interventions. Especially, the removal of lactose (galactose and glucose disaccharide), sucrose (glucose and fructose disaccharide), starch (a polymeric carbohydrate of glucose), sulfate, asparagine, and glutamine had a particular strong effect on multiple of the target metabolites (Fig. 5).

In addition, the doubling interventions involving nitrite, ubiquinol-8 (energy production), glycocholate, and pyruvate had desired effects on the fluxes of multiple target metabolites (Fig. 5).

## Discussion

In this study, we comprehensively investigated associations between host as well as microbial metabolic activities and disease activity in IBD and how patients differ by response to treatment and remission during treatment. Our modeling approach confirmed known disease-associated metabolic changes observed in the gut, such as the reduced production of SCFAs[24,25], the reduction of deconjugation of bile acids[26,28,29], and reduced tryptophan and histidine levels in patient serum[30-32]. Crucially, our metabolic modeling approach for the first time allowed us to link shifts in host physiology to changes in microbial metabolism and to investigate the interplay between different metabolic pathways affected during IBD-induced inflammation.

In our analysis of microbial metabolism, we found that overall metabolic activity was suppressed in inflammation. Essential cofactors like NAD, flavins, and tetrapyrroles seemed to be increasingly sourced from the diet rather than synthesized by the microbiome (Fig. 1B). It is known that microbially-produced vitamins significantly contribute to host vitamin metabolism[33] and deficiencies are associated with detrimental phenotypes[34,35] including IBD[36] and exacerbate inflammation[37]. Further, cross-feeding of amino acids, carbohydrates, and their degradation products among microbes was diminished (Fig. 1C). Consequently, we found that more amino acids became available to the host, while our models predicted a reduction of colonic carbohydrates and degradation products (including SCFA) due to altered microbial metabolism (Fig. 1D). Many of these changes have been described before[11,12,36,38-40] confirming our analysis. Clostridia and Bacilli, both belonging to the Bacillota phylum (formerly Firmicutes), appeared to contribute substantially to cross-feeding and host-exchanged metabolites (Fig. 1E, F). In line with our predictions, Clostridia abundances are decreased in IBD patients[39,41].

For host metabolism, we found a general reduction of metabolic activity with active disease and identified NAD, tryptophan, lipid metabolism, and general amino acid metabolism as key affected pathways, compatible with reduced serum metabolite levels involved in these pathways during inflammation in our and other studies[42,43]. At the same time, the changes in the microbial metabolism integrated well with and potentially contributed to host metabolic differences—especially in NAD and lipid metabolism. On the metabolomics level, we observed increased sphingolipid and fatty acid levels while phosphatidylcholines were reduced during active disease, directly reflecting the metabolic changes in the lipid metabolism derived from metabolic modeling. Furthermore, we found tryptophan serum levels decreased during active disease, in line with earlier studies[44], corroborating the

findings for tryptophan metabolism in the host. We confirmed these initial intuitions of highly interdependency of host and microbial metabolism and metabolite serum levels with a network and shortest-path analysis.

Additionally, we found NAD at the center of metabolic changes associated with disease activity in both tissues (Fig. 2B and Supplementary Fig. 5). Indeed, NAD metabolism and tissue levels have been described as relevant for inflammation in IBD[44]. We detected an increase in NAD degradation during inflammation, while the final reactions of salvage and de novo NAD synthesis were less active (Fig. 6), in line with previous reports[45]. NAD degradation is mediated by sirtuins and poly(ADP-ribose) polymerase (PARP) proteins and both are directly involved in the regulation of inflammatory responses[46]. NAD levels quickly diminish if the salvage is inactive[47] and decreased levels have been reported in inflamed tissue of UC patients[48,49]. The NAD salvage pathway is important for monocytes to induce a proper immune response[50] and invasion of activated monocytes into the inflamed tissues could explain our prediction of increased nicotinamide phosphoribosyltransferase activity. Furthermore, increased NAD salvage has been observed in treatment-responsive patients and the pathway has been proposed as a potential target for IBD therapy[45,51,52]. De novo NAD synthesis is fueled by the levels of quinolinate and its nicotinate derivatives. The microbiome is an important factor in regulating NAD levels in the gut by production of quinolate from aspartate[53], and we predicted decreased flux for this pathway in patients with high inflammation (Figs. 6A, B, and 1B). At the same time, quinolinate production by tryptophan degradation was suppressed, as indicated by the increased production of indole in the microbiome during inflammation (Fig. 1D). Interestingly, indole is an anti-inflammatory agonist of the aryl hydrocarbon receptor and has been reported as beneficial during colonic inflammation[54-57]. Tryptophan supplementation induces microbial indole production[58], which has been proposed to be a factor to establish and maintain host tolerance to microbe dwelling[59] and might allow less beneficial bacteria to colonize during inflammation. Furthermore, we found decreased tryptophan levels in the serum during inflammation (Fig. 3A) consistent with previous reports[29,30,49], accompanied by the prediction of increased degradation of tryptophan via the kynurenine pathway in biopsy samples (Supplementary Fig. 15A, B). The observed induction of the kynurenine degradation pathway is characteristic of inflammatory processes, as its products xanthurenate and kynurenate play crucial roles in reducing the inflammatory response[31,60]. Simultaneously, we predicted a blocked quinolinate production from kynurenate and consequently a reduced production of NAD from tryptophan (Supplementary Fig. 15A, B). Thus, the enrichment of inflammation-associated reactions using NAD as a substrate in blood and biopsy was likely a result of reduced NAD levels in the tissue, a consequence of increased NAD degradation, reduced de novo synthesis driven by changes in the microbiome and reduced tryptophan metabolism in the gut. This most likely led to enhanced inflammatory responses and can significantly contribute to disease progression, as shown in mice[37]. Further, our findings are supported by an inflammation-independent increase of microbial production of NAD and nicotinamide, an increased utilization of NAD in the gut, as well as increased levels of kynurenine and serotonin (indicating higher tryptophan degradation) in responders/remitters.

We predicted that the microbiome is producing more metabolites carrying an amino group during inflammation, increasing the provision to the patients. Yet, we predicted reduced activity of transamination reactions in patients and reduced urea cycle activity, indicating a demand for amino-groups. We speculated that the main sink for amino groups is the production of proinflammatory NO from arginine and indirectly glutamate in the gut (Supplementary Fig. 16C, D). This notion was supported by other studies where serum levels of amino

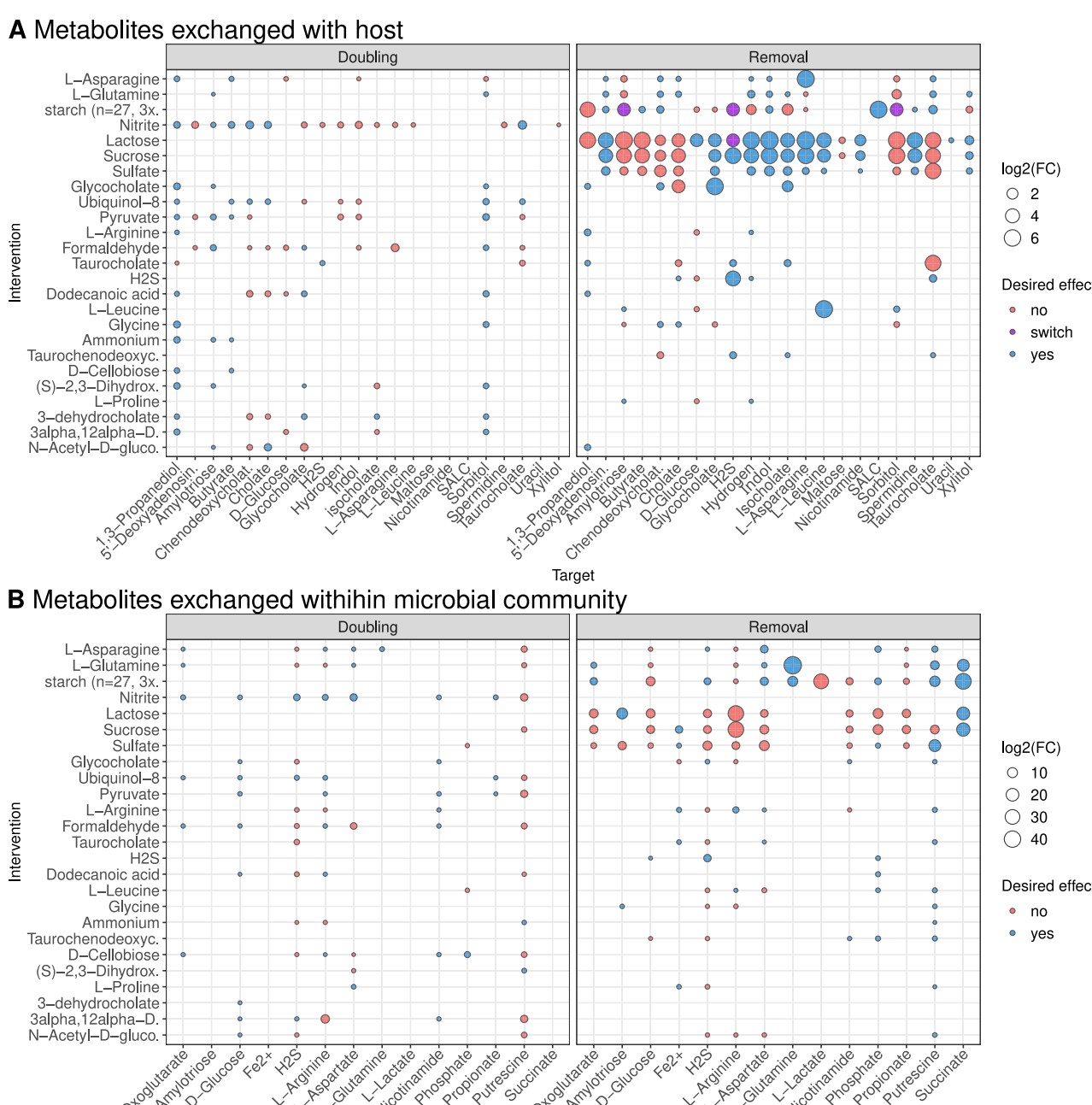

**Fig. 5 | In-silico interventions predict potential dietary therapies. A** The effects of doubling or removing dietary metabolites on the exchange fluxes of inflammation-associated metabolites between the host and the microbiome. The size of the circle reflects the intervention effect size (log2 fold change). The color of the circle represents whether the intervention is predicted to have inflammation-reducing effects. Desired influence in blue, (increased flux of metabolites negatively associated with inflammation or vice versa), undesired influence in red (increased flux of metabolites positively associated with inflammation or vice versa) and purple circles represent interventions that cause the target metabolite to change the direction of the flux (from the microbiome to the host before the intervention and from the diet to the microbiome after the intervention or vice versa). **B** Same as in Fig. 4A, but with the influence on metabolic fluxes observed for metabolic exchange within the microbial community that are associated with inflammation. **A, B** All interventions show mixed desired and undesired effects, with more positive influence of the removal of saccharides on the exchange between the microbiome and host and a negative influence on the exchange fluxes among the microbiome. Multiple testing adjustments via Benjamini–Hochberg correction. FC fold change.

acids were found to be decreased during inflammation, while amino acids glutamine and glutamate concentrations were increased in inflamed gut tissue[42,43]. NO is important for proper T-cell[61] and macrophage function as well as wound healing[62], while glutamine is an important suppressor of proinflammatory signals[63,64] and improves epithelial barrier function[65–67]. Further, host NO has been reported to be an electron acceptor in the microbiome[68], which might explain the

increased production of amino acids by bacteria to fuel NO production of the host for their own benefit. This hinted at a disturbance in nitrogen and amino acid levels with direct implications for the immune system. Indeed, glutamine/glutamate levels have been in the focus of several studies of IBD before[69,70], and arginine/glutamate supplementations have been tested as an IBD therapy in animals[71,72] and humans[73], with varying success.

We observed one-carbon metabolism as a hub connecting several of our findings including changes in lipid and NAD metabolism—with S-adenosylmethionine (SAM) at its core. We predicted a reduction of one-carbon metabolism in blood samples and a decreased production of homocysteine (precursor of SAM) by the microbiome (Fig. 6C, D). Homocysteine itself is also a direct modulator of immune system

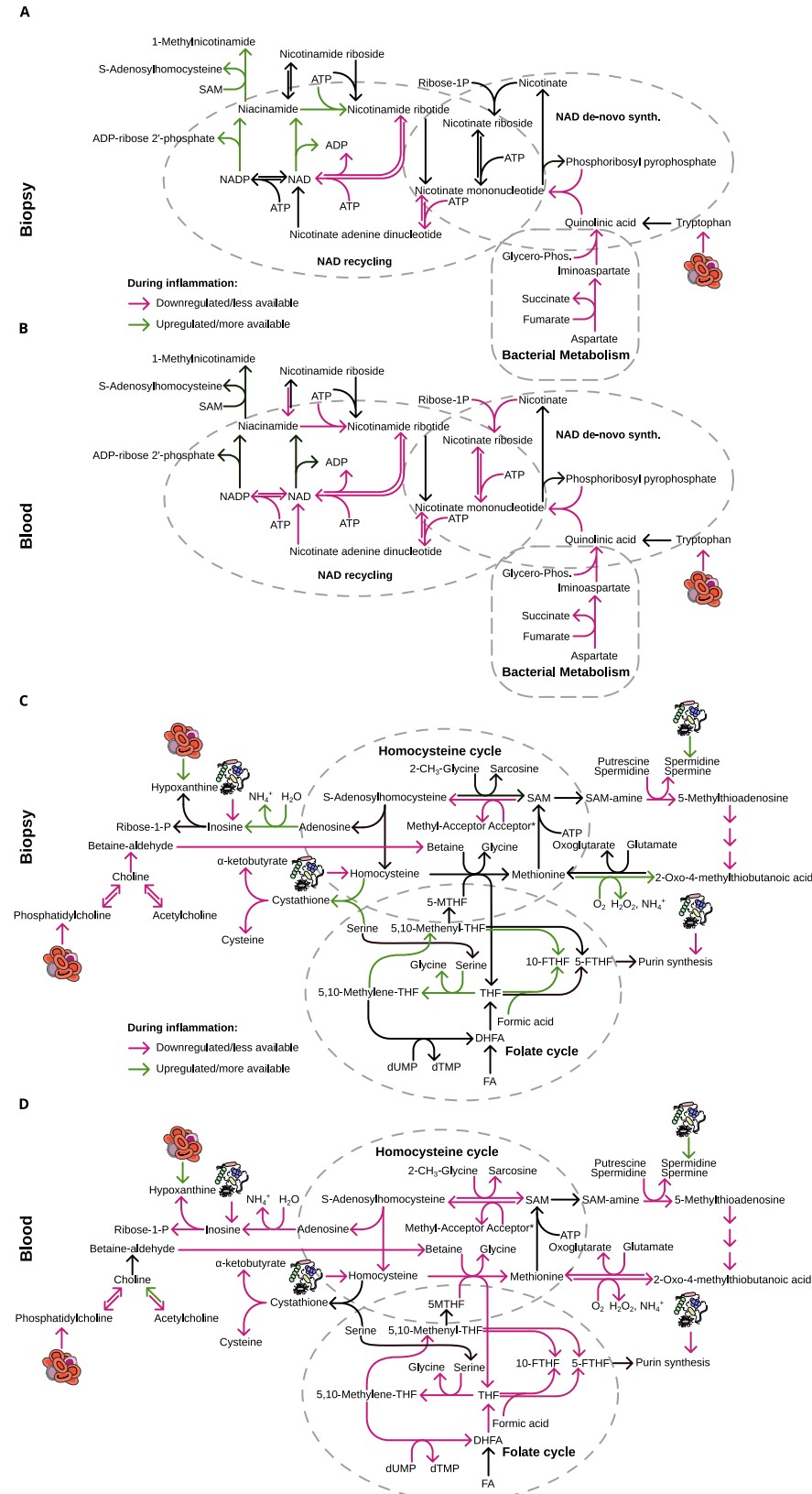

**Fig. 6 | Metabolic pathways connecting inflammation-associated reactions.** Schematic representation of inflammation-associated changes in NAD metabolism and associated pathways in biopsy (**A**) and blood (**B**). Accordingly, one-carbon metabolism and its associated pathways in biopsy (**C**) and blood (**D**).

functions and T-cell differentiation[74–76]. Next to folate metabolism, choline-derived betaine can serve as a donor of carbon to recycle homocysteine. Choline, an essential nutrient[77], is predominantly used for homocysteine recycling by the degradation of phosphatidylcholines. Our predictions showed reduced activity during inflammation in reactions liberating choline in both tissues (Fig. 6C, D). This is supported by decreased serum levels of phosphatidylcholines during inflammation (Fig. 3A, B) and, at the same time, might cause the observed shift from phosphatidylcholines to ceramides and glycosphingolipids in blood metabolomics. Glycosphingolipids fulfill similar functions in membranes as phosphatidylcholines and might be used to compensate for the loss of phosphatidylcholines due to choline usage in SAM recycling. Clinical and mouse studies have proposed phosphatidylcholine supplementation to treat gut inflammation, with positive effects on inflammation severity and various metabolic pathways[78–82]. Conversely, the knockout of the rate-limiting enzyme of phosphatidylcholine synthesis in mice causes severe colitis[83] and high sphingolipid levels have been described as pro-inflammatory[84–86]. Our data supports the notion that this proinflammatory shift in lipids is at least partly driven by the increased degradation of choline to recover homocysteine and C1 metabolism. Methyl-nicotinamide is an anti-inflammatory derivative of NAD and its synthesis is a major regulator of the substrate levels of SAM and NAD in the cell (Fig. 6A, B)[87–90]. We found increased activity of these reactions during inflammation in biopsies, which would further contribute to the lack of NAD and SAM during inflammation.

Finally, our dietary intervention simulation yielded predictions of metabolites that could be provided as adjuvant therapy to reestablish host-microbe and microbe-microbe interactions deregulated in IBD. Often, the same intervention had a desirable effect on microbiome-host exchanges but an undesirable effect on the exchanges within the microbiome in our models. This indicates that there is no one-fits-it-all solution for dietary interventions across all IBD-associated metabolic changes in the microbiome and the heterogenous microbial profiles in IBD patients. Furthermore our results imply that single metabolite supplementations are most likely not suitable as a general adjuvant therapy in IBD. Consequently, it also means that microbial heterogeneity complicates finding effective complex dietary interventions for IBD patients and requires highly individualized dietary plans which are tailored to the metabolism of the patients and their microbiome. Thus, it is less surprising that clinical studies which evaluate different dietary regimes and supplementations showed varying success[73,91]. Metabolic modeling could help identify these personalized diets by modeling the metabolism of the microbiome and the host and should be used as a supporting tool in future studies. Among the interventions, sucrose removal had mostly a desirable effect on the fluxes of metabolites associated with inflammation, which agrees with sucrose intervention experiments performed on mice with dextran sulfate sodium induced colitis[92,93]. Additionally, the positive effect of in-silico removal of polysaccharides and lactose are backed by studies that showed improvement in clinical remission after polysaccharides reduction and increased risk of lactose intolerance after lactose supplementation in IBD patients[94,95]. Other promising predictions were the removal of sulfate and the addition of nitrate. Increased sulfur levels lead to mucin degradation and contribute to colon inflammation in animal models[96] while administration of nitrate was found to increase colon length and alleviate existing colonic inflammation[97]. Further studies are needed to validate the effects of these metabolites through controlled dietary manipulations within bacterial communities, which could yield valuable insights into their role in reducing inflammation as potential adjuvant therapies.

Interestingly, when analyzing our data for markers of therapy response/remission, we found that responders/remitters are highly heterogeneous and that there is an effect of treatment (strength) on this metabolic heterogeneity. We were able to correct for some of this heterogeneity by accounting for disease activity, as a potential proxy for the treatment effects on response/remission. This finally led to the identification of metabolic functions, which are generally associated with response/remission, regardless of inflammatory state. In the future, this approach might be more useful to identify factors for treatment success which are independent of the disease state of IBD patients. Further, this might allow to disentangle molecular markers for wanted and unwanted treatment effects, which would help to identify markers for treatment response.

Furthermore, we would suggest some improvements for data collection in future studies. We used 16S amplicon sequencing and bulk RNA sequencing in order to infer high level interactions between the microbiome and the tissue of patients, lacking the finegrained resolution of strain and cell-specific information. Yet, modern metagenomic and single cell sequencing could unravel details of strain and cell-specific communication holding potential treatment avenues which are highly specific on these interactions.

With this study, we show that IBD is associated with a loss of microbiome and host metabolic activity. This disruption extends beyond the gut, showing a strong correlation between changes in directly affected tissue and those in the bloodstream. This tight interplay underscores the indispensable role of the microbiota in influencing host metabolic processes. We revealed that NAD, amino acid, one-carbon, and phospholipid metabolic pathways are intricately linked, and severely affected by intestinal inflammation. We argue that adjuvant dietary treatments should integratively consider these metabolic alterations and target all pathways simultaneously to obtain the maximal effect. Conversely, this might explain why certain dietary interventions had no effect in previous studies[73,91]. Simultaneously, our results enable the development of individualized therapies, considering the microbial as well as the host metabolism and their interplay. However, interventions targeting these pathways pose significant challenges as no predicted intervention demonstrated exclusively positive benefits. This highlights the complexity of IBD and the inadequacy of single-metabolite interventions and likely explains the failure of previous intervention approaches. It underscores the necessity for more comprehensive therapeutic strategies that account for the intricate interactions between host and microbiota.

## Methods
### Cohort description
To understand how metabolism in the microbiome and the host changes, we re-analyzed data from two previously published longitudinal IBD cohorts from northern Germany (trial IDs: EudraCT number 2016-000205-36 and ClinicalTrials.gov NCT02694588)[11,98,99]. These included a total of 62 patients, diagnosed with either CD or UC and treated with either anti-TNFα agents or with olamkicept (antagonizing IL6, Supplementary Fig. 1A). Patients were monitored for clinical and biochemical parameters, while samples of feces, blood and gut biopsy were taken at several time points over the course of 14 weeks. Inflammation states for both cohorts were estimated using the Mayo scoring system (Mayo) for UC and Harvey-Bradshaw index (HBI) for CD. These were monitored over time and used to define remission and response in the patients after treatment—which we did according to the original publications[11,98,99] (Supplementary Fig. 1B). To analyze both scores together, we censored the HBI values at a value of 16 to ensure similar scales for both disease activity scores. We imputed missing values for HBI/Mayo by a linear regression model for each individual patient over time. Within the two cohorts, 11 patients were defined as "healthy controls" as these showed no active inflammation at baseline or throughout the study (Supplementary Fig. 1B). The cohorts were overall balanced for different covariates, like sex, gender, therapy, and age (Supplementary Fig. 1B, C). Both cohorts were

analyzed simultaneously as we did not observe fundamental differences in disease activity changes over time and between diseases (Supplementary Fig. 17). Since our dataset lacked dietary information of patients, we estimated the nutrient availability by a data set of a similar cohort from northern Germany[100] (called Matjes diet, Supplementary Data 15).

### Metabolic modeling

**Community models.** The fecal 16S sequences were mapped against a set of 5416 human gut bacteria (the HRGM collection[18]) using a BLASTn algorithm. The mapping revealed the presence of 991 bacterial strains within the cohort. These strains were used to calculate abundances for each individual sample (Supplementary Data 16) and used to derive taxonomic and other information (Supplementary Data 17). We thereafter used gapseq[101] for the reconstruction of genomic scale metabolic models for each of the identified gut bacterial strains using the earlier described standardized diet, those models are compatible with the sybil R package model formats. These models were employed in the respective community modeling approaches. For some key statistics on the model size, number of genera in the model, number of uptake/secretion reactions and cross-feeding reactions refer to Supplementary Fig. 2 and Supplementary Data 18.

**Constraint-based modeling, coupling (MicrobiomeGS2).** For each of the microbial communities that were collected from the fecal samples, we reconstructed a community metabolic model using a coupling approach with MicrobiomeGS2 R package[11]. First, we excluded bacterial models with low estimated growth rates ($<10^{-3}$), as they can drastically reduce the overall growth of the community (in total 26 of 991 models, Supplementary Data 19) We then joined the bacterial models belonging to the same community in a merged model, where each model has its own compartment and all the models are connected through an environmental compartment to freely exchange metabolites. The microbial abundances (Supplementary Datas 16 and 17) were used as weighting factors for the communities biomass function, ensuring overall biomass production in the same proportion as observed in the experimental data. We set a coupling factor $c = 200$ which couples reaction flux of an organism to biomass production if the flux is larger than a coupling threshold $u = 0.01$ mmol $g_{Dw}^{-1}hr^{-1}$ as previously described[102]. CplexAPI R package[103] was thereafter used to run the metabolic simulations using the CPLEX solver with an academic license[104].

The simulations resulted in 245 metabolites being exchanged between the community and the host or among the community, or in both. After filtering out metabolites that have no exchange or the same exchange rate among all samples (no variance between the different communities), we ended up with 111 metabolites being exchanged among the community and 106 metabolites exchanged between the community and the host, 31 of which are produced by the communities, 60 are consumed by the communities, and 15 metabolites that are produced in some of the communities and consumed by others.

**Agent-based modeling (BacArena).** Besides flux coupling, we used another community modeling approach focusing on the independent optimization of microbial biomass production. In particular, we employed the individual-based community modeling framework BacArena (v.1.8.2)[19]. The initial abundance of bacterial models was set similarly to the coupling approach above. The spatial setup of the arena environment was $100 \times 100$ grid cells. In addition to the described standardized diet, concentrations of 1 mM for chorismate, indole, and salciate were assumed to ensure the initial growth of all models. The simulation was performed for three time steps, and afterwards, exchange fluxes for all samples were obtained. CplexAPI R package[103] was used to run the metabolic simulations using the CPLEX solver with an academic license[104].

**Host exchange, microbiome exchange, and overall flux.** To assess metabolic exchange of the bacteria with each other and with the host, we calculated the 'microbial exchange' and 'host exchange' values for each exchange metabolite in our community models. That is, for 'microbial exchange' the sum of all absolute fluxes for internal exchange reactions was subtracted by the overall exchange flux. Internal exchange reactions are those reactions which mark the model boundaries for each individual species within the microbial community and would be ordinary exchange reactions if each bacteria in the community would be modeled individually. For 'host exchanges' we simply used the exchange reaction fluxes for the whole community in coupling-based modeling and summed all exchange reaction fluxes for all bacteria in the model in the agent-based approach. Overall flux through a reaction for the whole microbial community was determined as the total sum of individual fluxes through the respective reaction in each microbial model.

**Simulating metabolic dietary interventions.** We aimed to predict candidate dietary supplementation that could potentially reverse parts of the observed alterations in the microbial metabolic fluxes associated with inflammation. Therefore, we conducted dietary intervention simulations to assess their impact on the metabolic flux of target metabolites. The selection of target metabolites was based on their consistent direction of the association with the three inflammation proxies (HB-Mayo score, remission, and response, i.e., all positive or all negative). To execute these interventions, we manipulated the lower bounds of exchange reactions for the intervention metabolites in the community models. Three distinct types of interventions were employed: doubling interventions, wherein the original metabolite flux in each community was doubled; and removal interventions, resulting in the complete elimination of the metabolite from the diet. Each dietary metabolic element was subjected to the two types of intervention individually. Subsequently, we repeated the community model simulations following each intervention by each dietary element and observed the fluxes of the target metabolites between the microbiome and the host's diet on the one hand, and among the microbiome member species on the other hand.

To statistically evaluate the impact of the dietary interventions on the target metabolic fluxes, we conducted paired Wilcoxon tests that compared the original flux of each target metabolite with its flux after each intervention. We specifically considered interventions that resulted in significant changes in the metabolic flux of at least one target metabolite across the microbial communities. Thereafter, we quantified the fold change in metabolic flux following each intervention, by employing the formula:

$$mean\left(\log 2\left(\frac{flux\ after\ intervention}{original\ flux}\right)\right)$$

Only fold changes exceeding 5% were deemed significant.

**Reconstruction of context-specific models.** Context-specific models were constructed based on the bulk RNAseq data for colon biopsies and blood samples of the patients. To link gene expression values to the human metabolic model (recon3D[105]) we determined reaction expression values using the gene-reaction rules. Read counts were first transformed to TPM values and the boolean gene-reaction operators were translated to sum($x$) or min($x$) for OR and AND operators, respectively. To obtain core reactions, we thresholded resulting reaction expression values with local and global thresholds. Global thresholds were defined as the 10ths and 90ths percentile of all reaction expression values across the respective tissue as lower and upper bound, respectively. All reactions below the lower bound were considered inactive, all reactions above the upper bound were defined as

core reactions. All reactions between these two bounds were thresholded on the 50ths percentile of the local expression values for that reaction (that is, the reaction activity score for this reaction across all samples in the respective tissue). Reactions above the boundary were considered as core reactions. The resulting core reactions were submitted to FASTCORE[106] to reconstruct context-specific metabolic networks. After context-specific model reconstruction, we performed flux variability analysis to estimate metabolic capabilities of the models using the implementation of cobrapy[107]. The methods for this workflow were implemented in the python corpse package (https://github.com/porthmeus/corpse)[108], while the reconstruction pipeline was implemented in a snakemake workflow (https://github.com/Porthmeus/CSMGen_miTarget). Reaction activity score (reaction expression, rxnExpr), presence/absence (PA) in the model after FASTCORE reconstructions and a flux variability analysis (FVA, minimal and maximal values converted into ranges and center of the reaction flux, FVA.range, FVA.center) were submitted to statistical analysis as individual data layer. We built linear mixed models for each reaction with the rxnExpr, PA, range.FVA, and center.FVA as fixed effects, patient identity as random factor and disease activity (HB/Mayo) as dependent variable.

### Data preparation and statistical analysis

Each data layer in our data analysis was first filtered for features with near zero variance (caret::nearZeroVar v6.0-93[109]), scaled and centered (base::scale R 4.2.2[110]) and clustered. Clustering was based on Pearson's correlation coefficient $\left(d = 1 - \sqrt{\rho^2}\right)$ and was performed with DBSCAN (fpc::dbscan v2.2-9[111]) using a maximal distance of 0.1 and a minimal cluster size of 3. This resulted in many small clusters and only grouped those features with very high correlation (e.g., reactions lying consecutively in a pathway). Finally, we fitted (generalized) linear mixed models to the data. We excluded most covariates to avoid overfitting and those included were knowledge informed (see Supplementary Data 9 for details). For model fitting, we used the lme4 (v1.1-31)[112] and statistical tests on individual coefficients were performed using lmerTest (v1.1-31, $t$-statistics with Satterthwaite's estimation for degrees of freedom)[113]. Compliance to model assumptions were tested with diagnostic plots and manual inspection of the plots. For generalized linear mixed models, residuals were modeled using the DHARMa package (v0.4.6)[114]. Actual formulas to build the models are given in Supplementary Data 9. Resulting significant associations were enriched with hypergeometric tests or gene set enrichment (GSEA) as implemented in clusterProfiler (v4.6.0)[115]. GSEA was performed on effect sizes for the coefficient in the given dataset. For human reactions, we calculated the mean for each reaction on the effect size for the model across the three data layers (rxnExpr, PA, FVA). Host reactions were used to enrich subsystems (pathways) as annotated in recon3D[105] using both hypergeometric tests and GSEA. For identification of enriched metabolites, we performed only hypergeometric tests. For microbial reactions, we enriched BioCyc pathway annotation as given in the gapseq database[101] using hypergeometric tests and GSEA. Microbial pathways have been manually regrouped and renamed in cases of high overlap of reactions in the enriched pathways.

To calculate taxonomic contribution to metabolite production in our microbial analysis we fitted individual linear mixed models to the production/consumption rates of each of the metabolites identified in the previous analysis using taxonomic identity and other confounders as explanatory variable (flux ~ source + seqtype + taxonomy + (1| PatientID)). Here we used fluxes of internal exchange reactions (that is, the exchange reaction of the individual bacteria in the community modeling approach) to measure the contribution of each taxa to the overall exchange flux of each metabolite. Each taxonomic level was tested individually within this framework, where all unassigned and low abundant species (represented in fewer than 5 samples) were

grouped to "Other". Taxonomic annotation was derived from the original annotation provided by the HRGM reference[18]. Statistical testing was performed using a log-likelihood ratio test, followed by a post hoc test for taxonomic contribution to flux deviation from 0 (emmeans v1.8.2[116]).

For all other tests, we performed bootstrapping and compared observed values against randomly shuffled expectations values. For all tests, a significance level of $p < 0.05$ was adopted and $p$-values were corrected for multiple testing using the Benjamini–Hochberger correction.

### Network reconstruction

For network analysis, we used all reactions which showed significant association with one of the IBD-associated phenotypes as the basis for the reconstruction of one network per tissue and phenotype association. We reconstructed the network by translating the metabolites and reactions into nodes and edges, respectively. Edge distances were calculated as the log of the sum of node degree, similar to the description in ref. 27. For layouting, we used the Fruchterman-Reingold algorithm and we calculated path lengths using the Dijkstra algorithm. Shortest paths of interest were defined as paths with a distance between two compounds being smaller than the lower 0.05 percentile of all shortest paths in the dataset. Network calculations have been performed in igraph for R (v1.3.5)[117], plotting and interactive graphs have been created with ggplot2 (v3.4.4)[118], ggraph (v2.1.0)[119], and visNetwork (v2.1.2)[120].

### Plotting

General data handling and plotting was performed in R using data.-table (v1.14.8)[121], ggplot2 (v3.4.4)[118], cowplot (v1.1.1)[122] and RColor-Brewer (v1.1.3)[123]. Boxplots were drawn with the following specification: center line, median; box limits, upper and lower quartiles; whiskers, 1.5 times interquartile range; points, outliers. Error bars in other plots represent the confidence interval of the shown estimate.

### Reporting summary

Further information on research design is available in the Nature Portfolio Reporting Summary linked to this article.

## Data availability

The raw sequencing data are available in the GEO database under accession code GSE191328 (Cohort 1) and GSE171770 (Cohort 2). All other data generated in this study have been deposited in a Zenodo repository under https://doi.org/10.5281/zenodo.13759863[124].

## Code availability

All data, analysis script, final, and intermediate results are available through Zenodo (https://doi.org/10.5281/zenodo.13759863[124]), scripts only are published at github (https://github.com/Porthmeus/IBDMetabolicModeling)[125]. The pipeline for context-specific model reconstruction is available at github (https://github.com/Porthmeus/CSMGen_miTarget)[126].

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

## Acknowledgements

We acknowledge support by the German Research Foundation within the framework of the Excellence Cluster "Precision Medicine in Chronic Inflammation" (project code EXC2167) and the research group miTarget (project code FOR5042) to C.K., P.R., K.A, and S.S. Moreover, we acknowledge support by the German Ministry for Education and Research within the scope of e:Med iTREAT (project code 01ZX1902A) to C.K., P.R., and e:Med TRY-IBD (project code 01ZX1915A), EKFS Clinician Scientist Professorship (K.A., 2020_EKCS.11) and the Joachim Herz Stiftung to K.A. The project has received funding support from the Innovative Medicines Initiative 2 Joint Undertaking (JU) ImmUniverse under grant agreement No. 853995 and the EU Horizon Europe project PerPrev-CID Project ID 101156542. The original study with olamkicept (EudraCT 2016-000205-36) was supported as an investigator initiated trial by an unrestricted research grant from Ferring (to S.S./University Hospital Schleswig-Holstein). We used and adapted the following original artwork in some of the presented figures: "erythrocyte", "erythrocyte-1", "erythrocyte-2", "erythrocyte-4", "acidophile-erythroblast", "nk-cell", "antibody-10" by Servier (https://smart.servier.com, CC-BY 3.0), "Adenomatous Polyposis 01 Normal Colonic Epithelium" by DBCLS (https://togotv.dbcls.jp/en/pics.html, CC-BY 4.0), "syringe with blood" by va (CC0), "data-report" by hongxia (Public Domain), "Next Gen Sequencer" by Ryan Kissinger (Public Domain), "Homo sapiens sapiens" by Katy Lawler (katy.lawler@live.com, CC-BY 4.0). In addition, we acknowledge the usage of ChatGPT3.5 and Claude in the writing process of the manuscript. The

LLMs were used to rephrase, shorten and increase readability of the original text. All text generated with AI was thoroughly checked for correctness and edited where needed to express the original ideas.

## Author contributions

Conceptualization: J.T., A.S.K., C.T., J.Z. and C.K. Data acquisition: P.R., K.A., F.T. and S.S. Methodology: J.T., A.S.K. and J.Z. Statistical analysis: J.T. Visualization: J.T. and A.S.K. Writing - Original Draft: JT. Writing - Final version: J.T., A.S.K., C.T. and C.K. Review & Editing: All co-authors. Supervision: C.K. and J.T. Funding acquisition: C.K., P.R., K.A, and S.S.

## Funding

## Competing interests

S.S. received personal fees for lectures and consultancy from: AbbVie, Amgen, Arena, Biogen, Bristol Meyers Squibb, Celgene, Celltrion, Falk, Ferring, Fresenius Kabi, Galapagos, Gilead, IMAB, Janssen, Lilly, MSD, Mylan, Novartis, Pfizer, Protagonist, Provention Bio, Roche, Sandoz/Hexal, Shire, Takeda, and Ventyx. F.T. received speaker's fees from Abbvie, Bristol-Myers-Squibb, Celltrion Healthcare, Dr Falk Pharma, Eli Lilly, Ferring Pharmaceuticals, Janssen/J&J, Takeda, and funding from Sanofi/Regeneron. KA received consulting fee from Abbvie, J&J, Takeda, Immunowissen, Dr. Falk Pharma, Ferring Pharmaceuticals. All other authors declare no competing interests.

## Ethics approval

All research complied with relevant ethical regulations, and usage of human tissue material and transcriptome data was approved by the ethics committee of the Medical Faculty of Kiel University (A156/03, A124/14).
