## [Transparent Peer Review file · Nature Communications]

Metabolic modeling reveals a multi-level deregulation of host-microbiome metabolic networks in IBD

Corresponding Author: Dr Jan Taubenheim

Version 0:

Reviewer comments:

Reviewer #1

(Remarks to the Author)

We have read "Metabolic modeling reveals a multi-level deregulation of host-microbiome metabolic networks in IBD" by Taubenheim et al. This manuscript presents a complex analysis of metabolic changes in both the host and microbiome during IBD. The authors applied omics data and metabolic modelling tools to two existing datasets with extensive phenotypic profiling incorporating 62 patients with UC or CD. They found loss of metabolic activity in both inflamed tissue and systemically in the host tied to changes in the microbiome. For example, they found that IBD is associated with decreased metabolic activity of pathways with NAD, amino acid, one-carbon and phospholipids. The authors used serum metabolomics to confirm model-based predictions and to identify 18 significant metabolites with levels that were associated with disease activity. A total of 16 of these were involved in one or more disease-enriched host subsystems identified via metabolic modelling. The results of this work are novel and could lead to dietary interventions to counteract inflammation associated changes in the microbiota. While the results are certainly interesting and the methods represent a tour-de-force of mathematical modelling, there are some drawbacks and questions that arise from the methods applied that must be further detailed. Most importantly, more details are needed for the construction and use of metabolic models. No details are presented on the size or properties of the microbial or host models. These properties dramatically impact the simulated results and thus their properties are essential to detail (at least in the supplement). Also, the authors should mention some of the limitations and drawbacks in the methods and data types applied to their discussion section to provide a balanced view of the results presented. We detail our major and minor comments below.

Major points

1. This study suffers from two major assumptions that are unavoidable based on the data being analyzed. First, the study utilized 16S data for analysis of the microbiome. This provides limited resolution into the strain-level diversity of organisms in the microbiome. Second, the study relies on bulk RNA sequencing for host tissue analysis which averages gene expression across diverse cell types. These two simplifications could mask important cell or strain-specific changes important to understanding IBD pathogenesis. While the authors did the best they could with the data available, these two limitations should be acknowledged and explicitly discussed in the discussion section of the manuscript.
2. More details are needed regarding the community models constructed. It looks like 991 bacterial strains were identified based on mapping of 16S sequences to the 5416 human gut bacteria represented in the HRGM collection. Were all of the 991 strains present in all of the samples? The authors mention different abundances but no differences in content between samples. The list of identified strains should be made available.
3. The methods section on constraint-based modeling and coupling describes distinct microbial communities derived from each of the fecal samples. More details are needed on the content of these communities. For example, the authors state that they excluded bacterial models with low estimated growth rates ($<10^{-3}$). How many models were excluded at this step for each sample?
4. Another supplementary figure describing the experimental design, community modelling setup and key metrics (e.g. model sizes, organisms in community, # of input/output metabolites, etc.) would be helpful to better summarize all of the interacting omics and model types. Something similar to Supplemental Figure 4 but with more details about each of the steps and sizes of the networks.
5. The use of shortest paths to examine the relationship between microbial and host metabolism needs more justification. It strikes me as a rather simplistic way to explore the question the authors are interested in particularly following the intricate and complex metabolic modelling. Path length tells us nothing about activity nor use of different pathways. Was the path length analysis performed on each patient specific microbiome community model? These details don't appear to be

provided anywhere. Again, a flow chart detailing the analyses performed and methods used would help the reader here.

Minor points

1. Minor terminology issues e.g. gene-reaction association vs “Gene-Protein-Reaction” rule (GPR).
2. 16s vs 16S
3. The MS doesn't have line numbers so minor comments are difficult to call out directly. Please review for spelling and grammar errors.

(Remarks on code availability)

We appreciate that the code has been placed into a github repository and appears to be well documented

Reviewer #2

(Remarks to the Author)

Brief Summary:

The authors used genome-scale metabolic models of both gut microbes and host tissue along with linear mixed models to identify metabolic changes that are associated with inflammatory bowel diseases. They considered both inter-microbiome and host-microbiome interactions. They observed multiple pathways that are up-/down-regulated in a disease state and saw consistency between different layers of data (microbe models, host models, and metabolomic data). Lastly, they predicted the effects of several potential dietary interventions on metabolic interactions in the gut.

Overall, some aspects of the paper are novel and have the potential for making an impact in the field. The analyses and interpretations are a valuable effort to identify specific metabolic pathways that are associated with inflammatory bowel diseases and potential avenues for dietary interventions. However, the paper needs to better communicate its approaches and results in the context of other work in the field and the presentation of results needs significant improvement.

Major Comments:

The authors need to articulate the novelty of their work, particularly in results sections 2 and 3, in the context of other work in the field. Two examples include Scoville et al, 2017, *Metabolomics* and Santoru et al., 2021, *Inflammatory Bowel Diseases*.

For each statistical analysis, please indicate what significance threshold (alpha) was used and what each star (*) represents i.e. $p < 0.05$.

Results section 1: There seems to be little agreement between the BacArena predictions and the MicrobiomeGS2 predictions (notably in Figure 1A). The lack of overlap should be addressed in the text.

Further clarify methods used in Figure 1C-F. For example, how were estimates averaged between bacteria (1C, E, F) and how is this different than the exchange data between host-microbiome (1D)? How was the contribution of each taxonomic group toward IBD-associated metabolites determined (1E, F)?

For Figure 2, in the methods it sounds like a separate LMM was created for each reaction. If so, how was a correction made for multiple hypothesis testing? Additionally, how was a reaction determined to be associated with disease activity (significance of overall correlation, significance of individual coefficients, etc.)? Furthermore, introduction of LMMs before the results would be beneficial.

Many of the figures are incredibly difficult to understand. For example, in Figure 2C it is difficult to decipher how to interpret the respective distributions, percentages, and ratios.

In Figure 4, the authors discuss how they find many closely related lipid reactions, but how many TOTAL “lipid reactions” are there? Is this number statistically significant?

The analysis on dietary interventions is insightful, especially as it highlights the challenge in effectively treating IBD with a single-metabolite intervention. Further discussion could be included about future efforts to identify dietary interventions either in silico or in vitro given the mixed results found here.

Minor Comments:

Figure 1 listed as Figure 2 in first paragraph.

Explicitly define “host exchange” and “microbial exchange”.

What does “ES” stand for in Figure 1E-F and what do error bars represent in Figure 1C-D?

Please restructure second to last sentence on page 8: "Secondly, similar metabolic ... and arachidonic/eicosanoid metabolism in microbiome and patients, respectively (Fig. 1B, Fig. 2A)." It is unclear what respectively is in reference to here.

What is the meaning of the purple "Switch" color in Figure 5A?

In the last sentence of the "Constraint-based modeling, coupling (MicrobiomeGS2)" methods section, you say "...15 are either produced in some of the communities and produced by others." Is one of these "produced" supposed to be "consumed"?

Figure text size could be increased in all figures to improve readability.

Supplemental table 4 and Table 1 can't be accessed (permission denied for the linked google doc).

All the supplemental tables should be standalone documents, shared with the journal rather than requiring access to a Google doc under the control of the authors.

(Remarks on code availability)

It appears to be well-documented. I did not independently run the code.

Version 1:

Reviewer comments:

Reviewer #1

(Remarks to the Author)

We thank the authors for their detailed response to our suggestions. We feel the manuscript is much stronger now and ready for publication

(Remarks on code availability)

Reviewer #2

(Remarks to the Author)

The authors well addressed the previous critiques. It is a nice contribution.

(Remarks on code availability)

Response to the reviewers

We thank the reviewers for their time and effort to read and improve the manuscript. The comments were very constructive and we found them very helpful. Our response to the reviewer comments follow in a point-by-point manner. The initial points raised by the reviewer are in black font, while our answer is written in dark blue for easier discriminability.

Reviewer #1 (Remarks to the Author):

We have read “Metabolic modeling reveals a multi-level deregulation of host-microbiome metabolic networks in IBD” by Taubenheim et al. This manuscript presents a complex analysis of metabolic changes in both the host and microbiome during IBD. The authors applied omics data and metabolic modelling tools to two existing datasets with extensive phenotypic profiling incorporating 62 patients with UC or CD. They found loss of metabolic activity in both inflamed tissue and systemically in the host tied to changes in the microbiome. For example, they found that IBD is associated with decreased metabolic activity of pathways with NAD, amino acid, one-carbon and phospholipids. The authors used serum metabolomics to confirm model-based predictions and to identify 18 significant metabolites with levels that were associated with disease activity. A total of 16 of these were involved in one or more disease-enriched host subsystems identified via metabolic modelling. The results of this work are novel and could lead to dietary interventions to counteract inflammation associated changes in the microbiota. While the results are certainly interesting and the methods represent a tour-de-force of mathematical modelling, there are some drawbacks and questions that arise from the methods applied that must be further detailed. Most importantly, more details are needed for the construction and use of metabolic models. No details are presented on the size or properties of the microbial or host models. These properties dramatically impact the simulated results and thus their properties are essential to detail (at least in the supplement). Also, the authors should mention some of the limitations and drawbacks in the methods and data types applied to their discussion section to provide a balanced view of the results presented. We detail our major and minor comments below.

Major points

1. This study suffers from two major assumptions that are unavoidable based on the data being analyzed. First, the study utilized 16S data for analysis of the microbiome. This provides limited resolution into the strain-level diversity of organisms in the microbiome. Second, the study relies on bulk RNA sequencing for host tissue analysis which averages gene expression across diverse cell types. These two simplifications could mask important cell or strain-specific changes important to understanding IBD pathogenesis. While the authors did the best they could with the data available, these two limitations should be acknowledged and explicitly discussed in the discussion section of the manuscript. Thank you very much for this comment. You are absolutely right about these shortcomings, which we did not discuss adequately in the current manuscript. We corrected this and added a short paragraph in the discussion addressing the advantage of using single cell and metagenomic sequencing in line 659-664.
2. More details are needed regarding the community models constructed. It looks like 991 bacterial strains were identified based on mapping of 16S sequences to the 5416

human gut bacteria represented in the HRGM collection. Were all of the 991 strains present in all of the samples? The authors mention different abundances but no differences in content between samples. The list of identified strains should be made available.

Thank you very much for pointing this out. We included the abundance table and the meta-data for the strain information as supplementary tables (S6 and S7) to the manuscript and described them in the methods section (lines 712-714 and 717-720). Not all strains could be detected across all samples, and community size was 50.1 ± 21.27 species per sample on average. We also provide a general overview of the models now in supplementary figure 2 and supplementary table 8.

3. The methods section on constraint-based modeling and coupling describes distinct microbial communities derived from each of the fecal samples. More details are needed on the content of these communities. For example, the authors state that they excluded bacterial models with low estimated growth rates ($<10^{-3}$). How many models were excluded at this step for each sample?

We thank the reviewer for this comment. 26 bacterial models out of 991 had estimated growth rates lower than the threshold 0.001 and were consequently removed before the community modeling. We added the information in the methods (line 725) and included a list of the models which failed to grow as a supplementary table 9.

4. Another supplementary figure describing the experimental design, community modelling setup and key metrics (e.g. model sizes, organisms in community, # of input/output metabolites, etc.) would be helpful to better summarize all of the interacting omics and model types. Something similar to Supplemental Figure 4 but with more details about each of the steps and sizes of the networks.

Thank you very much for this suggestion. We included a new panel in Supplementary figure 3 describing the general workflow of our analysis. The experimental design of the clinical trial was included in Supplementary figure 1. It is a good idea to check model sizes, and we added a supplementary figure summarizing the model statistics for the host and the microbiome in Supplementary Figure 2 and Supplementary Table 8. Furthermore, we referenced key numbers in the text in lines 135-136, 182-183, and 717-728.

5. The use of shortest paths to examine the relationship between microbial and host metabolism needs more justification. It strikes me as a rather simplistic way to explore the question the authors are interested in particularly following the intricate and complex metabolic modelling. Path length tells us nothing about activity nor use of different pathways. Was the path length analysis performed on each patient specific microbiome community model? These details don't appear to be provided anywhere. Again, a flow chart detailing the analyses performed and methods used would help the reader here.

Thank you for the comment. You are right, the method is not adequately described, which we corrected in the results (line 306-314) and the methods sections (line 860-862 and 864). In short, we used the significantly IBD-associated reactions of the human metabolic network to reconstruct a simple network with metabolites as nodes and reactions as edges. Hence, we removed the hypergraph properties, and further, we removed the compartment information and transport reactions to simplify the resulting network. We treated both tissues (blood and biopsies) separately, resulting in networks which are disease and tissue-specific across all samples in the data (one can consider it the metabolic consensus network of IBD in gut and

blood). Using only significant reactions for network reconstruction results in subnetworks that cannot necessarily carry flux anymore, and hence metabolic modeling approaches cannot be applied to these networks. Thus, the resulting networks were used to calculate shortest paths between the metabolites of interest using the shortest path lengths as described in the manuscript. Shortest paths in this context approximate pathways through the network as alternatives to elementary flux modes as described in Croes et al. 2005 (doi:10.1093/nar/gki437). Though the method is only an approximation with some room for errors, we would not expect a systematic error which would prefer shorter paths between metabolites of interest (IBD associated) and any other metabolite in the network, hence, we believe the method to be valid in our setting. We also added the reference in the methods and the main text of the manuscript. However, to add some more information, we made use of a recently published mouse metamodel (Best et al. 2025, doi:10.1038/s41564-025-01959-z) (including both microbial models and host models for different tissues, including the colon) to further confirm our findings. We used an elementary flux mode (EFM) sampling approach and counted how often each colon host reaction is sharing an EFM with microbial reactions. Elementary flux modes correspond to mathematically defined minimal pathways in a metabolic network (https://doi.org/10.1002/biot.201200269). This approach was applied to 52 context-specific mouse-microbiome meta-models, and we then calculated the mean co-occurrence across these 52 samples. We used the data to ask whether the IBD-associated microbial reactions are more likely to use common EFMs with IBD-associated host reactions. Indeed, we found that the fraction of EFMs going through host reactions more often contained microbial reactions if both reactions were IBD-associated, as opposed to reactions which were not IBD-associated (see attached figure). This confirms our conclusion that the metabolic changes in the host and microbiome due to inflammation are co-dependent. However, though confirmative, the model is based on mouse data, including many assumptions which are not necessarily correct for the human IBD data presented here. Hence, we decided not to include it in the current manuscript.

Minor points

1. Minor terminology issues e.g. gene-reaction association vs “Gene-Protein-Reaction” rule (GPR).

Thanks, this was fixed.

2. 16s vs 16S

Corrected.

3. The MS doesn't have line numbers so minor comments are difficult to call out directly. Please review for spelling and grammar errors.

Thanks for pointing this out. The new version of the manuscript contains line numbers and we read the manuscript once more for grammar and spelling.

Reviewer #1 (Remarks on code availability):

We appreciate that the code has been placed into a github repository and appears to be well documented

Reviewer #2 (Remarks to the Author):

Brief Summary:

The authors used genome-scale metabolic models of both gut microbes and host tissue along with linear mixed models to identify metabolic changes that are associated with inflammatory bowel diseases. They considered both inter-microbiome and host-microbiome interactions. They observed multiple pathways that are up-/down-regulated in a disease state and saw consistency between different layers of data (microbe models, host models, and metabolomic data). Lastly, they predicted the effects of several potential dietary interventions on metabolic interactions in the gut.

Overall, some aspects of the paper are novel and have the potential for making an impact in the field. The analyses and interpretations are a valuable effort to identify specific metabolic pathways that are associated with inflammatory bowel diseases and potential avenues for dietary interventions. However, the paper needs to better communicate its approaches and results in the context of other work in the field and the presentation of results needs significant improvement.

Major Comments:

1. The authors need to articulate the novelty of their work, particularly in results sections 2 and 3, in the context of other work in the field. Two examples include Scoville et al, 2017, Metabolomics and Santoru et al., 2021, Inflammatory Bowel Diseases. Thank you very much for this comment. We highlighted the novelty of our work now in more detail in the introduction (line 84-89), where we stressed the holistic approach of the modeling. Additionally, we emphasize that modeling provides mechanistic insights into causes of changes in metabolic functions, which can be exploited therapeutically. We discussed the two suggested metabolomics papers in the discussion (line 521-522 and 585-587) - thank you for your recommendation. We mention both papers now in regard to our findings of general reduction of metabolism, especially in NAD, lipid, and amino acid metabolism, which is confirmed by their analysis. Secondly, we mention common findings in our modeling approach of reduced amino acid metabolism and their finding of reduced amino acid levels in

IBD patients. Discussing novelty in specific result sections seemed inappropriate to us, especially given that the reconstruction of context-specific models and targeted blood metabolomics are not novel techniques on their own. The novelty of the study lies in the integration of diverse data layers and the functional interpretation of them in a holistic approach, as we have highlighted now.

2. For each statistical analysis, please indicate what significance threshold (alpha) was used and what each star (*) represents i.e. $p < 0.05$.

Thank you for noticing this omission. For all tests, we assumed a $p < 0.05$ as statistically significant and all analyses were corrected for multiple testing with the Benjamini-Hochberger adjustment. We added this information in the respective method section (line 857-858) and exchanged the asterisk in the figures with the actual p-value as well as stated the statistical test in the figure caption where applicable (line 301-302).

3. Results section 1: There seems to be little agreement between the BacArena predictions and the MicrobiomeGS2 predictions (notably in Figure 1A). The lack of overlap should be addressed in the text.

Thank you very much for the comment. The two approaches model different aspects of microbial ecology. While MicrobiomeGS2 optimizes community growth, hence emphasising cooperation and cross-feeding between bacterial members, which would optimize overall growth of the community, BacArena optimizes individual models, emphasizing competition for nutrients and only allows for the evaluation of by-product cross-feeding. To assess all aspects of microbial metabolism, we included both approaches in our analysis. This also explains differences in the detection of IBD-associated changes in the microbial metabolism, which agrees in the majority of cases where both methods detected the same metabolites. We now explain this in more detail in the text in lines 143-145. Furthermore, both modeling approaches show similarity in flux prediction. We have correlated the flux prediction for both methods across samples and for each reaction to show this. While most reactions are not significantly correlated, among those that are, the majority is positively associated. We attached the figure showing these results here for reference.

4. Further clarify methods used in Figure 1C-F. For example, how were estimates averaged between bacteria (1C, E, F) and how is this different than the exchange data between host-microbiome (1D)? How was the contribution of each taxonomic group toward IBD-associated metabolites determined (1E, F)?

Thank you very much for these questions. We did not average estimates across bacteria, but used the sum of all reaction fluxes through the individual reactions in each bacteria as a measure for the overall flux in the microbial community (Fig 1B). For microbial exchange we took the sum of absolute values of fluxes of the internal exchange reactions (that is the exchange reaction of the individual microbial model in the community) and removed the absolute value of the model exchange reactions (so the actual exchange reaction, characterizing the uptake/secretion of the microbial community from/to the environment) - this denotes the flux which is occurring between the bacteria. For the host exchange, we simply used the overall exchange flux with the environment for the coupling-based approach or the sum across all metabolite exchanges for the agent-based modeling. We explain these details now in a new subsection in the methods (line 755-766). The calculation of the taxonomic contribution was done on the individual exchange fluxes of each bacterium. We used these values in each sample and checked whether the microbial contribution to the overall flux was statistically different from 0. The statistical approach is detailed in the methods section for data handling and statistics. We expanded this part of the section with the information on how fluxes from the microbes were derived (line 846-849).

5. For Figure 2, In the methods it sounds like a separate LMM was created for each reaction. If so, how was a correction made for multiple hypothesis testing?

Additionally, how was a reaction determined to be associated with disease activity (significance of overall correlation, significance of individual coefficients, etc.)? Furthermore, introduction of LMMs before the results would be beneficial.

Thank you for pointing out the missing information in the description of our methods. Indeed, we constructed LMMs for each reaction, which we made more clear now in line 192-193. Furthermore, we used t-statistics after Satterthwaite's estimation of degrees of freedom to assess the relevant coefficient in the LMM for statistical significance and multiple testing was always regarded by Benjamini-Hochberger correction - which we now clearly indicate in the method section (line 828 and 857-858).

6. Many of the figures are incredibly difficult to understand. For example, in Figure 2C it is difficult to decipher how to interpret the respective distributions, percentages, and ratios.

Thank you for this advice. We adjusted the labels of some of our plots to increase clarity and we changed Figure 2C, by splitting it into two separate plots (Fig. 2C, D). The first shows the reaction activity of the reactions employing the hub metabolites as a boxplot (similar to the boxplot in Fig. 2A), and a second plot shows the proportion of reactions where the metabolite serves as substrate or product. Further, we adjusted the figure caption accordingly and changed the references in the text.

7. In Figure 4, the authors discuss how they find many closely related lipid reactions, but how many TOTAL “lipid reactions” are there? Is this number statistically significant?

Thank you for this comment, but we are a little unsure what “TOTAL lipid reactions” refers to, so we decided to enrich subsystems with the reactions along the shortest paths analysis described in Figure 4. We found a significant enrichment for glycerophospholipids and sphingolipid metabolism as described. Furthermore, we could confirm the enrichment of amino acid metabolism with this analysis, including methionine and cysteine metabolism, which comprises the C1 metabolism. To show these results, we added supplementary figure 6 and described the results in lines 350-352 and 374-377. We hope this addresses the concern adequately.

8. The analysis on dietary interventions is insightful, especially as it highlights the challenge in effectively treating IBD with a single-metabolite intervention. Further discussion could be included about future efforts to identify dietary interventions either in silico or in vitro given the mixed results found here.

Thank you very much for this recommendation. We included a new part in the discussion where we touch on the difficulty of defining dietary interventions and that there is no silver bullet dietary intervention (line 626-635).

Minor Comments:

1. Figure 1 listed as Figure 2 in first paragraph.

Thanks for spotting this - we corrected this mistake.

2. Explicitly define “host exchange” and “microbial exchange”.

Thanks - you are right, that was insufficiently defined. We added a small section in the methods (line 755-766) and added some reference in the results part to guide the reader.

3. What does “ES” stand for in Figure 1E-F and what do error bars represent in Figure 1C-D?

Thanks for the question - it stands for effect size, here the t-value for the statistics for the coefficient. Error bars indicate the confidence interval for the values. We added the information in the figure caption (line 125).

4. Please restructure second to last sentence on page 8: “Secondly, similar metabolic ... and arachidonic/eicosanoid metabolism in microbiome and patients, respectively (Fig. 1B, Fig. 2A).” It is unclear what respectively is in reference to here.

Thanks for spotting this. We removed a redundant part of the sentence, which hopefully makes more sense now (line 251).

5. What is the meaning of the purple “Switch” color in Figure 5A?

Thanks for pointing this out. We added to the Figure legend:

Purple circles represent interventions that cause the target metabolite to change the direction of the flux (from the microbiome to the host before the intervention and from the diet to the microbiome after the intervention or vice versa) (line 485-488)

6. In the last sentence of the “Constraint-based modeling, coupling (MicrobiomeGS2)” methods section, you say “...15 are either produced in some of the communities and produced by others.” Is one of these “produced” supposed to be “consumed”?
Thanks for noticing this mistake. You are right and we corrected this (lines 742-743).
7. Figure text size could be increased in all figures to improve readability.
Thank you for the comment. We adjusted the layout of our figures to better fit the pages and increased the font size to improve readability.
8. Supplemental table 4 and Table 1 can't be accessed (permission denied for the linked google doc). All the supplemental tables should be standalone documents, shared with the journal rather than requiring access to a Google doc under the control of the authors.
Thank you very much for the comment. We apologize for not making the tables accessible to the reviewers. We actually intended to submit all tables as csv files together with the manuscript and provided the links to google docs only for convenience. With this revision we made sure to upload all tables as additional csv files. Furthermore, we included table 1 now as supplementary table 6 and made Table S4 available for everybody on google docs.

Reviewer #2 (Remarks on code availability):

It appears to be well-documented. I did not independently run the code.

We thank the reviewer again for their time and effort to revise the manuscript and hope we have fully answered all questions and raised points.